# The effects of proteasomal inhibition on synaptic proteostasis

Vicky Hakim[1,2,†], Laurie D Cohen[1,2,†], Rina Zuchman[3], Tamar Ziv[3] & Noam E Ziv[1,2,*]

## Abstract

Synaptic function crucially depends on uninterrupted synthesis and degradation of synaptic proteins. While much has been learned on synaptic protein synthesis, little is known on the routes by which synaptic proteins are degraded. Here we systematically studied how inhibition of the ubiquitin-proteasome system (UPS) affects the degradation rates of thousands of neuronal and synaptic proteins. We identified a group of proteins, including several proteins related to glutamate receptor trafficking, whose degradation rates were significantly slowed by UPS inhibition. Unexpectedly, however, degradation rates of most synaptic proteins were not significantly affected. Interestingly, many of the differential effects of UPS inhibition were readily explained by a quantitative framework that considered known metabolic turnover rates for the same proteins. In contrast to the limited effects on protein degradation, UPS inhibition profoundly and preferentially suppressed the synthesis of a large number of synaptic proteins. Our findings point to the importance of the UPS in the degradation of certain synaptic proteins, yet indicate that under basal conditions most synaptic proteins might be degraded through alternative pathways.

**Keywords** proteasome; proteasome inhibitors; protein degradation; SILAC; synaptic proteostasis

**Subject Categories** Neuroscience; Post-translational Modifications, Proteolysis & Proteomics

The EMBO Journal (2016) 35: 2238–2262

## Introduction

Chemical synapses are sites of cell–cell contact specialized for transmitting signals between neurons and other neurons, muscles, or glands. As in nearly all biological structures, proteins in synapses have finite lifetimes and need to be continuously replaced with freshly synthesized copies. Synaptic protein homeostasis (proteostasis) presents daunting challenges from the neuron's perspective: The number of its synaptic sites can be huge, their distance from the cell body, where most protein synthesis occurs, is often enormous, and their makeup—membranal, cytoskeletal, and vesicular proteins—is extraordinarily complex. Historically, these challenges garnered considerable attention, leading to the eventual discovery of elaborate trafficking systems (reviewed in Maeder *et al*, 2014) and distributed capacities for protein synthesis (reviewed in Holt & Schuman, 2013). More recently, synaptic proteostasis has returned to the center of attention due to mounting evidence for ties between dysregulated synaptic proteostasis and a variety of neurodegenerative conditions (Keller *et al*, 2002; Bingol & Sheng, 2011; Yang *et al*, 2013; Buffington *et al*, 2014; Yamamoto & Yue, 2014; Ciechanover & Kwon, 2015).

While much has been learned on the synthesis and trafficking of synaptic proteins, less is known on the manners by which synaptic proteins are degraded. In this regard, degradation via the ubiquitin-proteasome system (UPS) has received much attention, with the accumulation of evidence that key synaptic proteins undergo ubiquitination and that suppressing proteasomal activity affects global or synaptic levels of synaptic proteins (Ageta *et al*, 2001; Chin *et al*, 2002; Wheeler *et al*, 2002; Ehlers, 2003; Colledge *et al*, 2003; Moriyoshi *et al*, 2004; van Roessel *et al*, 2004; Kato *et al*, 2005; Saliba *et al*, 2007; Yao *et al*, 2007; Jurd *et al*, 2008; Hung *et al*, 2010; Schwarz *et al*, 2010; Lazarevic *et al*, 2011; Tsai *et al*, 2012; Rezvani *et al*, 2012; Shin *et al*, 2012; reviewed in Bingol & Sheng, 2011; Schwarz & Patrick, 2012; Lin & Man, 2013; Tsai, 2014; Alvarez-Castelao & Schuman, 2015). While the evidence is compelling, it must be acknowledged that the overall contribution of UPS-based protein degradation to synaptic protein catabolism remains unclear. Thus, for example, protein ubiquitination is not necessarily indicative of imminent proteasomal-based degradation, as ubiquitination (in particular mono-ubiquitination) can act as a signal for downstream events or as an effector of protein function, without affecting protein levels (as shown for the presynaptic proteins Rim1 and Munc13; Rinetti & Schweizer, 2010; see also Bianchetta *et al*, 2011; Pinto *et al*, 2016). In addition, ubiquitination events are continuously countered by deubiquitinating enzymes. Consequently, the presence of conjugated ubiquitin (e.g. Na *et al*, 2012) cannot be taken as unequivocal evidence that a protein is destined for degradation (Udeshi *et al*, 2012). Finally, and perhaps most

1 The Rappaport Faculty of Medicine and Research Institute, Haifa, Israel
2 Network Biology Research Laboratories, Technion – Israel Institute of Technology, Haifa, Israel
3 Smoler Proteomics Center, Faculty of Biology, Technion, Haifa, Israel
*Corresponding author. Tel: 972 4 8295088; E-mail: noamz@netvision.net.il
†These authors contributed equally to this work
[The copyright line of this article was changed on 28 November 2016 after original online publication.]

importantly, much of the evidence concerning UPS involvement in synaptic protein catabolism is based on pharmacological agents that inhibit proteasomal activity; the interpretation of such experiments, however, is not straightforward, as the involvement of the UPS in myriad cellular processes can lead to unexpected results, including the suppression of protein synthesis (Ding *et al*, 2006, 2007a; King *et al*, 2008; Milner *et al*, 2013; Larance *et al*, 2013; see also Dong *et al*, 2008, 2014; Bajic *et al*, 2012; Jarome & Helmstetter, 2014). In fact, pharmacological agents that inhibit proteasomal activity often have rather modest effects on overall and synaptic levels of most synaptic proteins (e.g. Lazarevic *et al*, 2011; Shin *et al*, 2012). Thus, while cumulative evidence points to roles for UPS-based protein catabolism in synaptic proteostasis, the degree to which this pathway is involved in basal catabolism of most synaptic proteins remains unknown.

In the current study, we used multiplexed (Zhang *et al*, 2011, 2012) and pulsed (Selbach *et al*, 2008; Cambridge *et al*, 2011) SILAC (Stable Isotope Labeling with Amino acids in Cell culture) combined with MS (Mass Spectrometry; Pratt *et al*, 2002; Aebersold & Mann, 2003; Ong & Mann, 2006; Milner *et al*, 2006; Cox & Mann, 2008; Liao *et al*, 2008; Spellman *et al*, 2008; Doherty *et al*, 2009; Schwanhäusser *et al*, 2011) to systematically measure how the pharmacological suppression of proteasomal activity selectively affects the degradation (and synthesis) rates of thousands of neuronal and synaptic proteins. Using these approaches, we identified synaptic proteins whose degradation is strongly UPS dependent. We also found, however, that for the majority of synaptic proteins, degradation rates are not significantly slower in the presence of proteasomal inhibitors. In contrast to the modest effects on synaptic protein degradation, we show that UPS inhibition is associated with a profound suppression of synaptic protein synthesis. Finally, by synthesizing these data with information on metabolic turnover rates of individual proteins, we provide a quantitative framework which satisfactorily explains some of the differential effects of UPS inhibition yet also delineates limitations in the use of proteasomal inhibitors to study synaptic protein catabolism.

## Results

### Lactacystin strongly suppresses proteasome-mediated protein degradation in primary cultures of rat neurons

In the current study, we set out to examine to what degree basal synaptic protein catabolism is affected by the suppression of proteasomal activity. The pharmacological agent we selected to suppress UPS-mediated protein degradation in these (and subsequent) experiments was lactacystin. Lactacystin is a potent, highly selective proteasome inhibitor (Fenteany *et al*, 1994) whose main mode of action is the irreversible modification of the proteasome β5 subunit at its highly conserved active site, an amino-terminal threonine residue (Fenteany & Schreiber, 1998). Neither lactacystin nor its active intermediates detectably affect many other commonly tested proteases (Fenteany *et al*, 1995; Fenteany & Schreiber, 1998); moreover, and in contrast to the commonly used proteasome inhibitor MG-132, lactacystin does not inhibit calpains. Thus, lactacystin is considered to be a highly selective inhibitor, and when

used at a concentration of 10 μM, it has been shown to robustly block proteasome activity (e.g. Zhao *et al*, 2003; Willeumier *et al*, 2006; Machiya *et al*, 2010; Keyomarsi *et al*, 2011; Bajic *et al*, 2012).

To quantify proteasome inhibition in our preparations, we performed a series of experiments based on biochemical and imaging approaches. We started by measuring proteasomal activity in extracts of cortical neurons grown in culture for 2 weeks, treated (or untreated) prior to extraction with lactacystin (10 μM) for 4 h using the fluorogenic substrate N-Succinyl-Leu-Leu-Val-Tyr-7-amino-4-methylcoumarin (Suc-LLVY-AMC; Meng *et al*, 1999; Fornai *et al*, 2003; Kisselev *et al*, 2003; Kisselev & Goldberg, 2005). As shown in Fig EV1A and B, lactacystin pretreatment reduced Suc-LLVY-AMC fluorescence accumulation rates by a factor of ~20 (two separate experiments; see Materials and Methods for further details). We also quantified lactacystin effects on fluorescence accumulation in living cells using a similar fluorogenic substrate (LLVY-Rhodamine-110) and time-lapse imaging. Here too, lactacystin reduced fluorescence accumulation rates (~3-fold) although quantification was complicated by limited substrate penetration and availability as well as poor cytosolic retention of fluorescent products (Fig EV1D; 121 cells, four experiments).

Readouts based on fluorogenic substrates do not always provide accurate reports of proteasomal inhibition, due to the multiple proteolytic activities of proteasomes and important differences between short substrate degradation in extracts and protein degradation in cells (Dantuma *et al*, 2000). We thus measured how lactacystin affects the degradation of two proteins that are well-known proteasome targets. We first examined how lactacystin affects intracellular levels of p21[waf1/cip1] (cyclin-dependent kinase inhibitor 1A), a short-lived protein whose UPS-based degradation has been well characterized (Abbas & Dutta, 2009; see also Larance *et al*, 2013) and whose steady-state expression in cortical neurons is very low (Langley *et al*, 2008). As shown in Fig EV2, exposure to lactacystin greatly increased p21[waf1/cip1] levels over the course of 4–8 h. As a second approach, we used N-end rule degradation signal green fluorescent proteins (Dantuma *et al*, 2000). These proteins, in particular those destabilized by placing an arginine at position 1 (Ub-R-GFP), are rapidly targeted for UPS degradation. Consequently, when expressed in living cells, nominally low fluorescence levels increase substantially when UPS-mediated degradation is suppressed. Ub-R-GFP was expressed in cultured cortical neurons and followed by time-lapse imaging. As shown in Fig 1A and B, exposure to lactacystin (10 μM) led to dramatic increases in GFP fluorescence over timescales of hours. Unfortunately, because basal fluorescence was negligible, detecting Ub-R-GFP-expressing cells proved to be challenging. To overcome this difficulty, we co-expressed cyan fluorescent protein (CFP), allowing us to locate Ub-R-GFP-expressing neurons before applying lactacystin (Fig 1C). Furthermore, the coexpressed CFP allowed us to control for non-specific effects on cell morphology and for off-target effects on transcription (Biasini *et al*, 2004). As shown in Fig 1D, lactacystin induced dramatic increases in GFP fluorescence but no change in CFP fluorescence. Conversely, carrier solution application barely affected GFP fluorescence (Fig 1E). Comparing fluorescence change rates indicated that lactacystin increased GFP accumulation rates ~20-fold (Fig 1F), in good agreement with our measurements based on fluorogenic peptides (Fig EV1A and B). Taken together, these experiments

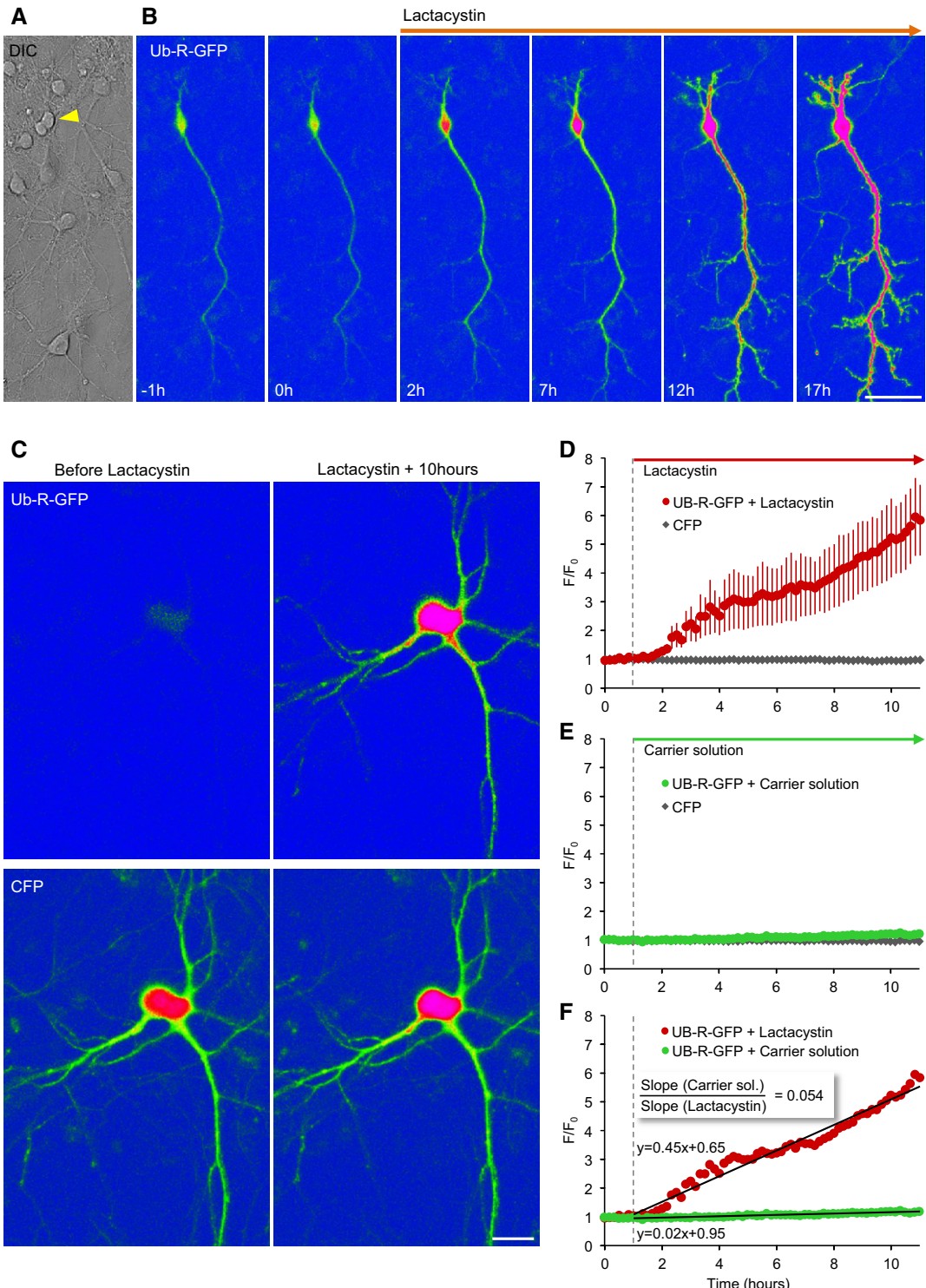

**Figure 1. Lactacystin strongly suppresses UPS-mediated protein degradation in living neurons.**

A   Differential interference contrast (DIC) image of cortical neurons in culture. The arrowhead points to a neuron expressing Ub-R-GFP.

B   GFP fluorescence before and after exposure to lactacystin (10 μM). Scale bar, 50 μm.

C   A neuron coexpressing Ub-R-GFP and CFP before (left) and 10 h after exposure to lactacystin (right). Top row, GFP fluorescence; bottom row, CFP fluorescence. Scale bar, 20 μm.

D   Changes in GFP and CFP fluorescence following exposure to lactacystin at $t = 1$ h (vertical dashed line). Changes normalized to initial fluorescence in same cells. Nine neurons from five experiments; averages and SEM.

E   Same as in (D), except that here neurons were exposed only to carrier solution. 11 neurons from five experiments; averages and SEM (barely observable).

F   Comparison of GFP fluorescence accumulation rates. Linear fits shown as black lines; fit parameters shown next to fits. Same data as in (D) and (E).

suggest that in the preparations used here, lactacystin suppresses proteasomal-mediated protein degradation by at least a factor of 10, probably more.

## Effects of proteasomal inhibition on synaptic levels of synaptic proteins measured by quantitative immunocytochemistry

Prior studies indicate that synaptic protein degradation might be, at least for some prominent examples (the presynaptic protein Rim1 and the postsynaptic proteins Shank and PSD-95; Colledge *et al*, 2003; Ehlers, 2003; Yao *et al*, 2007) mediated by the UPS (see Introduction). To measure the effects of lactacystin on synaptic levels of these and additional synaptic proteins, we used quantitative immunofluorescence to determine how 10-h exposures to lactacystin (10 μM) affect synaptic levels of RIM, ProSAP2/Shank3 and PSD-95, synapsin I, Bassoon, and SV2A. Hippocampal neurons were used here, as the lower densities of cells and synapses in these networks facilitated antibody access and single synapse quantification. As shown in Fig 2A and B, for most of these proteins, exposure to lactacystin was not associated with increases in their synaptic levels, as might have been expected from reduced degradation rates. If anything, slight reductions in synaptic levels of these proteins were observed. An exception was RIM, in which a small (19%) but highly statistically significant increase was observed ($P < 10^{-9}$).

A more detailed analysis was performed for Shank3/ProSAP2. Here, neurons were fixed at two-hour intervals following exposure to lactacystin. As shown in Fig 2C and D, no significant changes in the synaptic immunofluorescence levels were observed over a 24-h period (two experiments, 8–54 fields of view per time point).

As this limited survey did not provide strong evidence for widespread UPS-based synaptic protein catabolism (even though it was biased toward proteins previously reported to be degraded via the UPS), we set out to examine this question on a larger scale and in an unbiased fashion by SILAC and MS.

## Effects of proteasomal inhibition on apparent half-lives of synaptic proteins measured by pulsed SILAC

Metabolic half-lives of proteins are dictated by their synthesis and degradation rates. We reasoned that pharmacological inhibition of the proteasome might prolong the half-life of proteins whose degradation is primarily UPS mediated. To measure how proteasomal inhibition affects the half-lives of synaptic proteins, we adapted a dynamic SILAC approach we recently used to systematically measure synaptic protein half-lives (Cohen *et al*, 2013). In those experiments, lysine and arginine, containing non-radioactive, heavy isotopes of particular atoms ("heavy" amino acids), were added abruptly to rat cortical neurons raised in culture for 2–3 weeks, which were thereafter subjected to MS analysis at increasing times (days). The underlying logic was that proteins synthesized after exposure to heavy (H) amino acids (AAs) would contain mainly heavy lysine and arginine (Cambridge *et al*, 2011), whereas proteins synthesized prior to that time would contain only "light" (L) variants of these AAs. By quantifying changes in the ratios of heavy to light (H/L) peptides over time, the turnover rates of specific proteins could be estimated.

To estimate how half-lives of synaptic proteins are affected by pharmacological suppression of proteasomal activity, we raised

dense networks of rat cortical neurons in culture for 2 weeks in standard media, after which heavy lysine and arginine (Lys8 - $^{13}C_6$, $^{15}N_2$; Arg10 - $^{13}C_6$, $^{15}N_4$) were added in excess, resulting in a 5:1 H/L ratio of these AAs in the media. Lactacystin (10 μM) was added to one of the preparations, whereas carrier solution alone was added to the control preparations. Heavy AA addition, rather than replacement, allowed us to avoid the abrupt and complete media exchanges which can be rather strong perturbations in their own right (e.g. Saalfrank *et al*, 2015). Excess lysine and arginine, on the other hand, have no adverse effects on neuronal viability, synaptic density, spontaneous electrical activity levels, or protein expression profiles (Cohen *et al*, 2013). After 24 h, the neurons were lysed and extracted; the extracts were separated on polyacrylamide gels, which were subsequently cut into five sets of bands according to molecular weight. Each gel slice was then subjected to MS analysis, and an H/L ratio for each identified peptide was determined. H/L ratios from at least 2 unique peptides belonging to each protein were pooled, providing an average H/L ratio for each protein. The entire process is illustrated in Fig 3A.

Two separate experiments were performed, resulting in 1,409 proteins for which H/L ratios were obtained in both experiments and both conditions. As shown in Appendix Fig S1, H/L ratios (expressed as log₂H/L) for both the control and lactacystin data sets were highly repeatable (Pearson's correlation $r = 0.95$ and 0.91, respectively). We thus averaged data from both experiments and used these average H/L values in subsequent analyses. As shown in Fig 3B, comparisons of H/L values obtained from lactacystin-treated and untreated preparations suggested that proteasomal inhibition introduced major changes in H/L ratios, with the correlation dropping to 0.70. We then calculated, for each protein, the fold change in the H/L ratio [or more specifically, $\log_2(H/L)_{Lactacystin} - \log_2(H/L)_{Control}$] and sorted these from smallest to largest. As shown in Fig 4A, H/L ratios of most proteins were reduced (suggestive of slower turnover), although elevated H/L ratios were observed for some proteins (a matter we will return to later). Remarkably, when H/L ratio changes of 160 synaptic proteins were examined (see Materials and Methods for synaptic protein selection criteria), these tended to cluster at the leftmost region of this plot (greatest reductions in H/L ratios) which would seem to suggest that their turnover was slowed down dramatically. Thus, for example, within the group of proteins for which twofold or greater reductions in H/L ratios were observed, we found many well-characterized postsynaptic proteins (such as glutamate receptor subunits, CaMKIIα and β, PSD-95, Drebrin) and presynaptic proteins (Bassoon, synapsin I and II, syntaxin-1B, SNAP25, neurexin-2, synaptophysin, synaptotagmin-1, dynamin-1, synaptojanin-1, Munc18-1, NSF, Soluble NSF Attachment Protein; Fig 4, Table EV1). In fact, an unbiased gene ontology analysis (GORILLA: Gene Ontology enRIchment anaLysis and visuaLizAtion tool, http://cbl-gorilla.cs.technion.ac.il/; Eden *et al*, 2009) indicated that the "synapse part", "terminal bouton", and "synaptic vesicle" groups were the strongest affected subcellular groups ($P = 7.09 \times 10^{-10}$, $P = 1.41 \times 10^{-9}$ and $P = 5.91 \times 10^{-9}$, respectively, "Component", Fig EV3).

Under certain assumptions (discussed at length in Cohen *et al*, 2013), H/L ratios can be used to estimate half-lives of identified proteins. Indeed, half-life estimates based on the single 24-h time point of the control data set correlated well with prior estimates based on four time points (Cohen *et al*, 2013; $r = 0.75$; 1,335

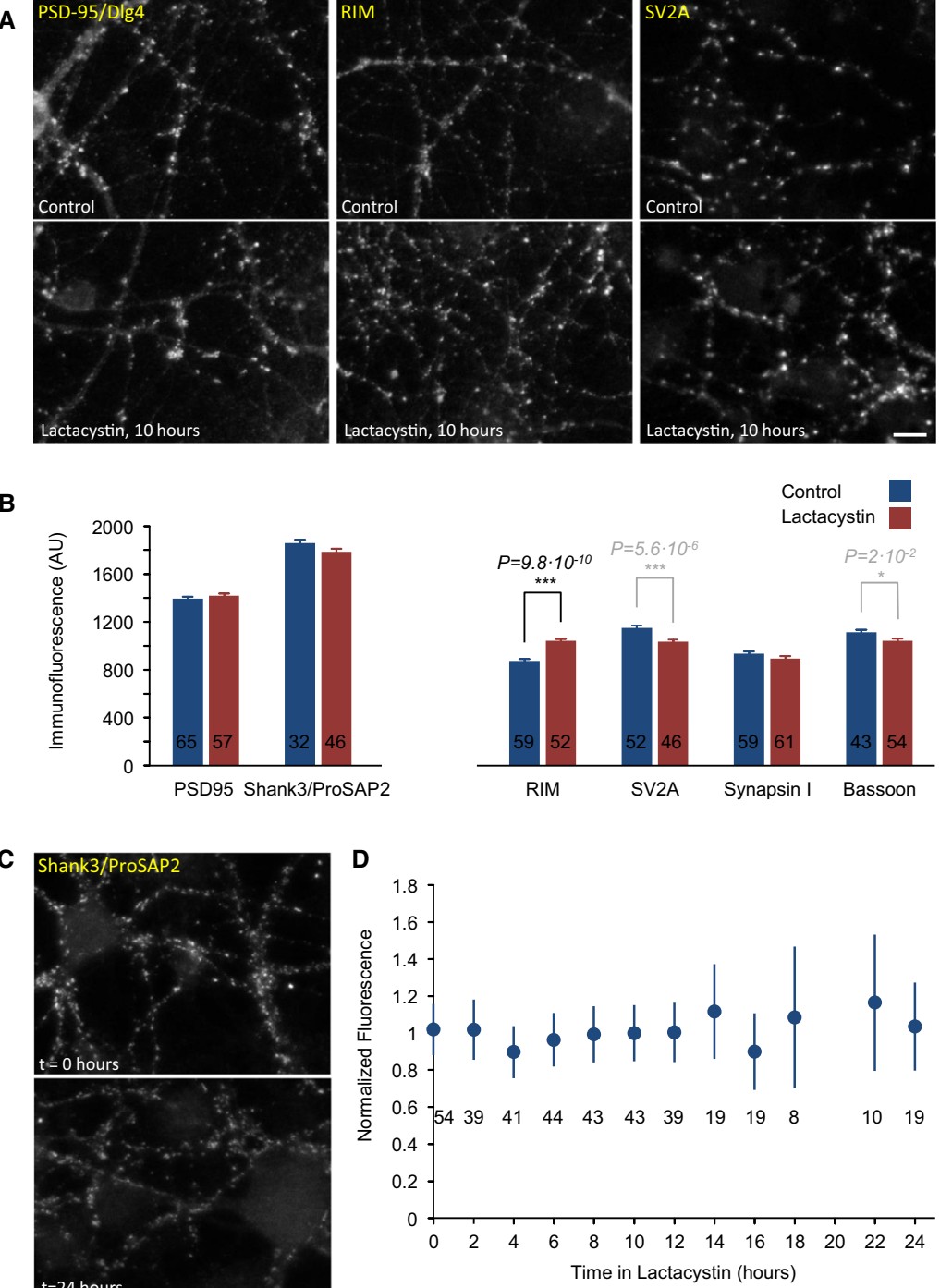

**Figure 2. Proteasomal inhibition does not lead to generalized accumulations of synaptic proteins at synaptic sites.**

A   Hippocampal neurons in primary culture immunolabeled against three synaptic proteins (the postsynaptic scaffold molecule Dlg4/PSD-95, the presynaptic active zone molecule RIM, and the presynaptic vesicle protein SV2A) following 10-h incubations in carrier solution (control, upper panels) or lactacystin (10 μM; bottom panels). Scale bar, 10 μm.

B   Average synaptic immunofluorescence following 10-h incubations in lactacystin or carrier solution. Numbers within bars represent the number of fields of view analyzed for each condition (~170 synapses/field of view, on average). Error bars, SEM; *P*-values calculated according to the two-tailed unpaired *t*-test. Note that a statistically significant accumulation was observed only for RIM, whereas immunolabeling of all other proteins was either unchanged or reduced.

C   Hippocampal neurons in primary culture were grown in the presence of lactacystin (10 μM), fixed at 2-h intervals, and immunolabeled against Shank3/ProSAP2. Two representative fields of view at *t* = 0 and *t* = 24 h. Scale bar, 10 μm.

D   Average synaptic Shank3/ProSAP2 immunofluorescence at increasing times in the presence of lactacystin. Fluorescence was normalized to mean fluorescence at *t* = 0. Numbers under data points reflect the numbers of fields of view analyzed at each time point. Error bars, SEM.

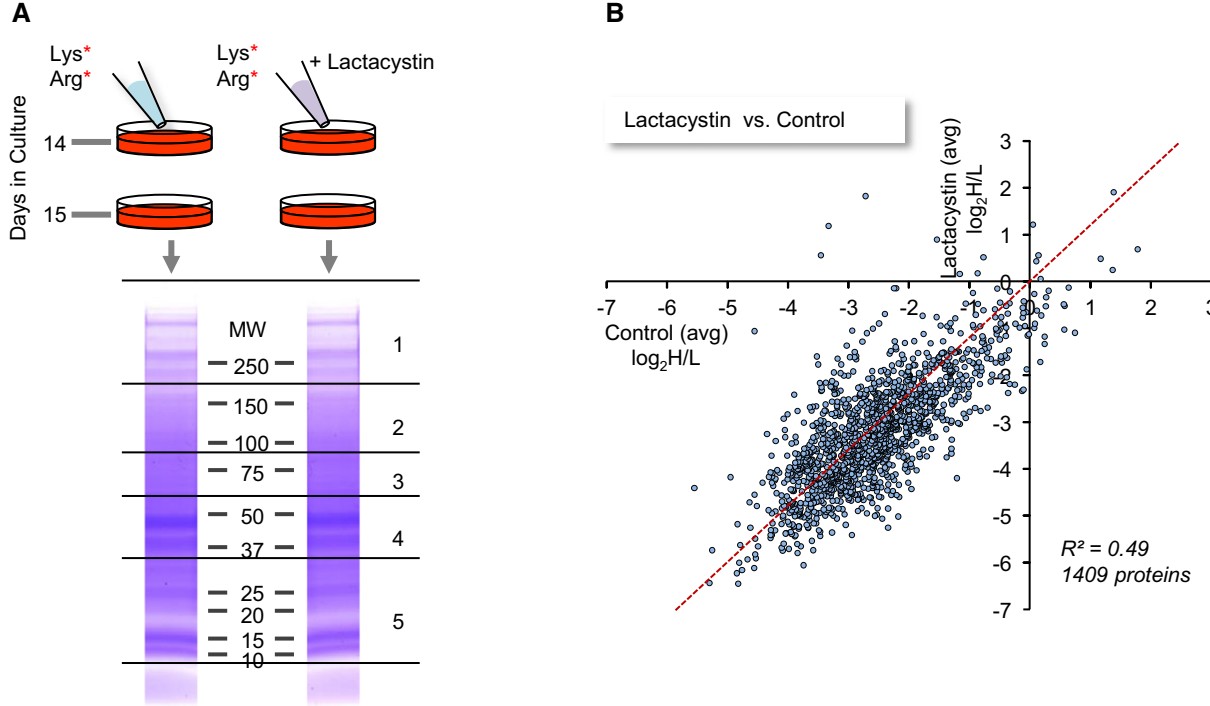

**Figure 3.  Measuring apparent changes in protein turnover by pulsed SILAC.**

A   Illustration of the experimental procedure. Following 2 weeks of growth in normal media (containing light variants of lysine and arginine), heavy lysine and arginine were added in excess as described in the Results section. Lactacystin was added to one of the sets at this time, resulting in a final concentration of 10 μM. 24 h later, cells were harvested and protein lysates were separated side by side by SDS–PAGE. Two lanes in such a gel (stained with Coomassie Blue) are shown. Lanes were then cut into five slices as indicated, proteins in each slice were digested, and the resulting peptides from each slice and each treatment were submitted separately to MS analysis, resulting in a list of H/L ratios for each peptide.

B   Comparisons of H/L ratios obtained for 1,409 proteins in lactacystin and untreated cells. H/L ratios are averages of two experiments. $R^2$ is the linear regression coefficient of determination.

proteins common to both data sets; Appendix Fig S2, Table EV2). Using the same method (see Materials and Methods), we estimated the apparent half-lives of the 1,409 proteins mentioned above in lactacystin-treated and untreated preparations, and plotted the apparent fold change in half-lives for synaptic proteins (Fig 4B). This analysis seemed to suggest that lactacystin prolonged the half-lives of most synaptic proteins by factors of ~2 to 7. However, as mentioned above, estimations of half-lives from H/L ratios, and consequently this interpretation, are based on certain assumptions, not all of which remain valid once the proteasome is inhibited. Of these, the most important is the assumption that protein synthesis and protein degradation rates are balanced, and thus, heavy AAs incorporation rates—which reflect protein synthesis—are mirror images of light AA loss rates—which reflect protein degradation. Once the proteasome is inhibited, however, this assumption is no longer valid; moreover, the basis for changes in H/L ratios becomes unclear: Do such changes reflect reduced protein degradation rates (larger denominators), or possibly, reductions in protein synthesis rates (i.e. smaller numerators)? The latter possibility cannot be ignored in light of prior reports concerning the effects of proteasome inhibitors on protein synthesis (see Introduction). Thus, a different approach was needed; one that would allow us to unequivocally measure the effects of proteasome inhibition on protein degradation rates and separate these from effects on protein synthesis.

**Using multiplexed dynamic SILAC to measure the effects of proteasomal inhibition on protein degradation rates**

To selectively examine the effects of proteasomal inhibition on synaptic protein degradation, we designed experiments for quantifying the rates at which *preexisting* protein pools are degraded, free from "contamination" by possible effects on protein synthesis. The experiments, based on a combination of multiplexed (Zhang *et al*, 2011, 2012) and pulsed SILAC, were performed as follows (Fig 5A): Cortical neurons were grown in culture for 2 weeks in the presence of isotopically separable lysine and arginine, that is, "heavy" (H) variants (Lys8 - $^{13}C_6$, $^{15}N_2$; Arg10 - $^{13}C_6$, $^{15}N_4$) or "medium" (M) variants (Lys6 - $^{13}C_6$; Arg6 - $^{13}C_6$). These AAs were added to lysine and arginine-free cell culture media such that total lysine and arginine concentrations were identical to nominal concentrations in our standard cell culture media. After 14 days *in vitro*, the media was replaced gently with media containing no isotopically labeled AAs and a 5× excess of light lysine and arginine. At this time point, lactacystin (10 μM) was added to the set of preparations labeled with heavy AAs. The cells were harvested after 4, 10, or 24 h, mixed together (as pairs of time-matched lactacystin-treated and control sets), run together on preparative gels, digested and subjected to MS analysis as detailed in Materials and Methods.

In this scheme, the peptide mixture subjected to MS analysis contained (i) unlabeled peptides originating in proteins synthesized

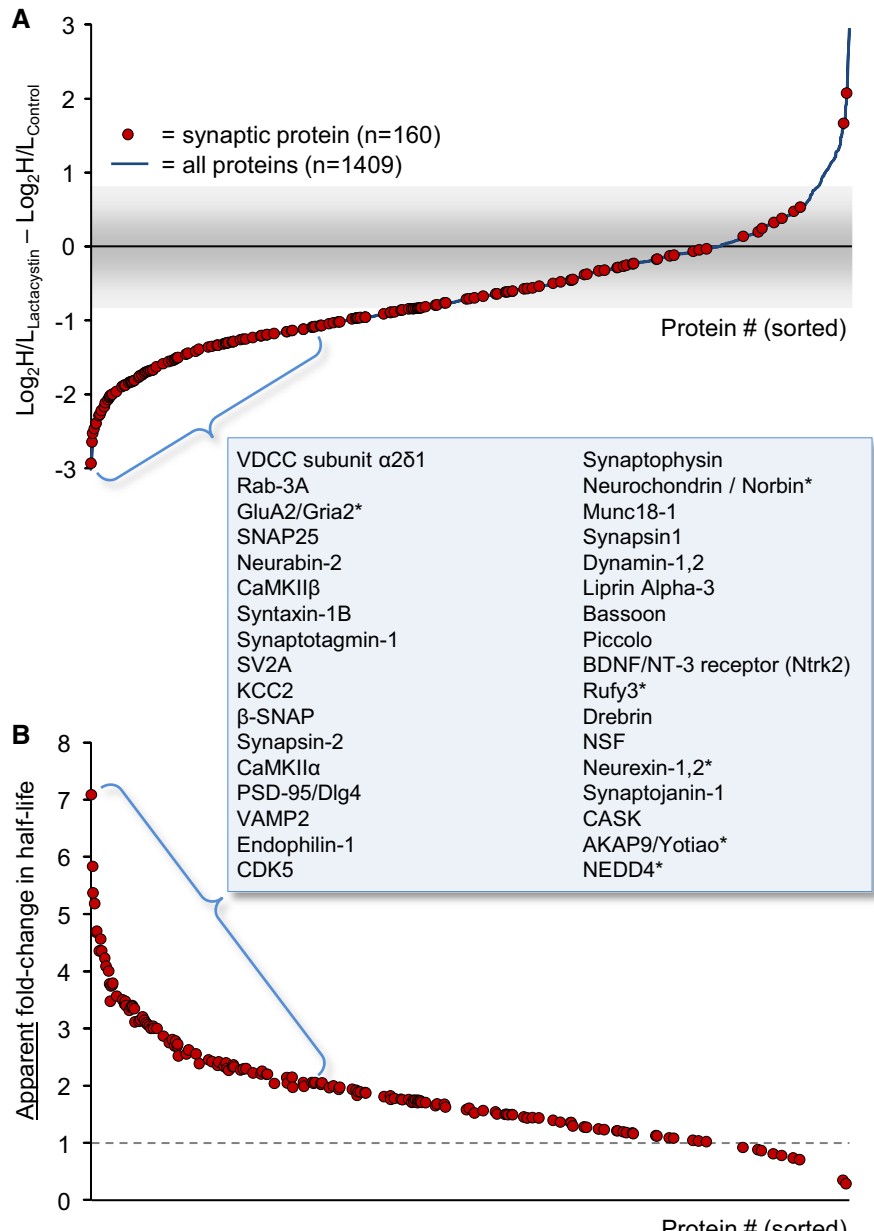

**Figure 4.   Lactacystin-induced changes in H/L ratios.**

A   Fold changes in the H/L ratios $\left(\frac{H/L_{Lactacystin}}{H/L_{control}}\right)$ measured for each protein in the experiments described in Fig 3 (expressed as $\log_2(H/L)_{Lactacystin} - \log_2(H/L)_{control}$). H/L ratios used here are the average of ratios obtained in two experiments for each protein and each condition. Proteins were sorted according to the fold change in H/L ratios from the proteins whose ratios were most strongly reduced (~8-fold reduction) to proteins whose H/L ratios increased to the greatest extent (> 8-fold increase). Red circles indicate the location of synaptic proteins along this sorted list. The names of 34 synaptic proteins located at the left-most region of the sorted list are provided in the order of their appearance.

B   The apparent fold change in protein half-lives under certain assumptions described in the Results section. See Materials and Methods for further details on the calculation of the apparent fold change in half-lives.

after the substitution of labeled AAs with unlabeled variants; and (ii) peptides labeled with either heavy or medium AAs, representing preexisting protein pools, that is, proteins synthesized prior to the removal of labeled AAs and exposure to lactacystin. The labeled peptides, in turn, were a mixture of peptides from the lactacystin-treated neurons (labeled with heavy AAs) and peptides from the untreated neurons (labeled with medium AAs). By calculating the

ratios of labeled peptide quantities for each peptide at each time point, the relative amounts of these peptides in treated and untreated cells could be determined. These ratios, referred to here as H/M ratios, (heavy/medium ratios, equivalent to experiment/control ratios) were then pooled for all peptides belonging to a particular protein, giving an average H/M ratio for each protein. These H/M ratios were then used to estimate the degree to which proteasomal inhibition slowed

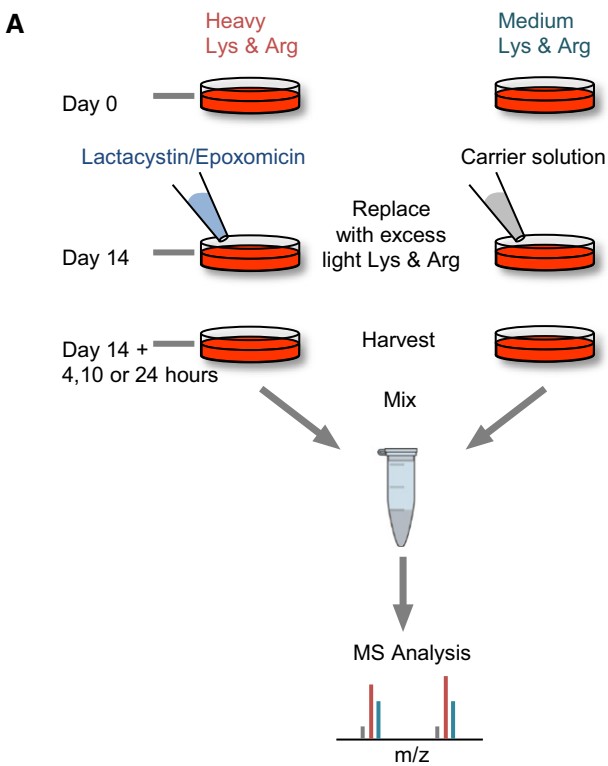

**Figure 5. Measuring changes in degradation rates by multiplexed SILAC.**

A  Illustration of the experimental procedure. Cells were grown for 2 weeks in growth media containing either heavy or medium variants of lysine and arginine. At day 14, the media was replaced with normal growth media containing an excess of unlabeled (light) lysine and arginine. Lactacystin or epoxomicin was added to one of the sets at this time. 4, 10, or 24 h afterward, cells were harvested, mixed, and separated by SDS–PAGE. Lanes were then cut into five slices, digested, and analyzed by MS analysis, resulting in a list of H/M ratios for each peptide.

B  Illustration of the meaning of H/M ratio. For proteins whose degradation is retarded by proteasomal inhibitors, H-labeled copies are degraded at slower rates than M-labeled copies, resulting in H/M ratios > 1. For proteins whose degradation is not affected by proteasomal inhibitors, H-labeled and M-labeled proteins are degraded at similar rates, resulting in H/M ratios of ~1.

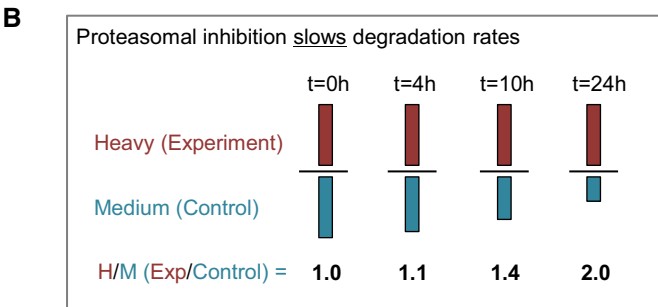

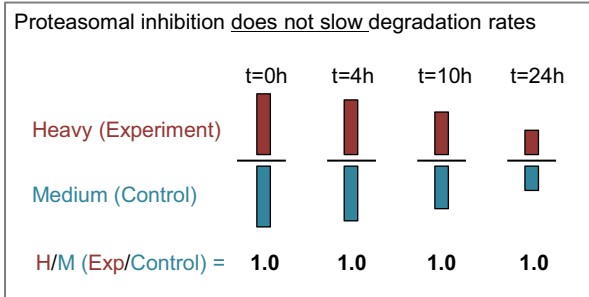

Importantly, because peptides containing unlabeled (light) AA variants were ignored in this scheme, estimates were insensitive to changes in amounts of newly synthesized proteins and were thus highly selective for effects on protein degradation. It is worth noting that the use of heavy and medium AAs was reversed in some of the experiments to control for nonspecific effects (see Materials and Methods), but for the sake of simplicity, we refer here and throughout the manuscript to experiment/control ratios as H/M ratios.

The validity of the scheme described above depends on a requirement that at the 14-day time point, protein pools in neurons growing in heavy or medium AAs are labeled to a similar degree and preferentially at levels close to saturation. This was examined experimentally by measuring the ratios of labeled to unlabeled peptides (H/L or M/L ratios) in neurons that had been grown in either heavy or medium AAs for 14 days (before lactacystin treatment). H/L and M/L ratios for all identified proteins were converted into fractional saturation values (i.e. $\frac{H}{H+L}$ and $\frac{M}{M+L}$; see Materials and Methods) and compared. As shown in Fig EV4, labeling was very similar and nearly complete and for both labels (91.8 and 91.1%, 1,134 and 922 proteins, respectively). However, these values were almost certainly underestimates, as no light peptides whatsoever were detected for most identified proteins (3,814 and 4,067 proteins, respectively) suggesting that for the vast majority of proteins, labeling was practically complete.

The fact that material extracted from treated and control sets was mixed, prepared, and analyzed together dramatically reduced the variability associated with multiple preparations and runs, as each peptide from the treated set was compared to an internal standard from the control set. Some method was still required, however, to correct for slight differences in the relative amounts of material collected from treated and control sets. To that end, we normalized all H/M (experiment/control) ratios to H/M ratios measured for a group of seven abundant and particularly long-lived proteins (Toyama *et al*, 2013; Table 1). The first criterion assured accurate quantification, while the second warranted that even if their degradation is UPS mediated, a 4- to 24-h-long suppression of proteasomal activity would have negligible effects on their quantities. All H/M ratios mentioned hereafter reflect H/M ratios normalized in this fashion.

Three separate experiments were performed in which a total of > 4,500 proteins were identified. Within this group, we selected 1,750 proteins for which (i) H/M (experiment/control) ratios were quantified for at least 2 peptides in each experiment, and (ii) such ratios were obtained in all three experiments at all four time points. H/M ratios for each protein were normalized to the ratio for the

the degradation of each protein, as illustrated in Fig 5B. Thus, for example, a H/M (experiment/control) ratio of ~1.0 for a certain protein at a particular time point indicated that the residual amounts of that protein were identical in treated and untreated neurons, and thus, its degradation was not slower in the presence of proteasomal inhibitors. Conversely, H/M ratios > 1.0 indicated that the residual amounts of that protein were greater in the treated cells, and thus, its degradation in the presence of proteasomal inhibitors was impeded.

**Table 1. Abundant long-lived proteins used for normalization.**

| Protein | $t_{1/2}$ (days) from Cohen *et al* (2013) |
|---|---|
| Vcan/Cspg2 | 9.64 |
| Lamin-B1 | 19.8 |
| Lamin-B2 | 16.1 |
| Nup155 | 9.37 |
| Nup205 | 14.09 |
| Macro-H2A.1 | 10.17 |
| Macro-H2A.2 | 10.62 |

The list is based on Toyama *et al* (2013); half-lives for the same proteins obtained in networks of cortical neurons in culture are from Cohen *et al* (2013).

aforementioned long-lived proteins for each experiment and time point (see Materials and Methods) and expressed as base 2 logarithmic values. As shown in Fig 6A, the average $\log_2(H/M)$ at $t = 0$ h for all 1,750 proteins after this normalization was very close to 0 ($-0.005 \pm 0.04$; $n = 3$); that is, initial amounts of identified proteins obtained from lactacystin-treated and control neurons were identical on average, as might be expected. To estimate the accuracy and sensitivity of our approach, we averaged the $\log_2(H/M)$ values obtained for each identified protein at $t = 0$ h and examined the distribution of these average values. As Fig 6B shows, average $\log_2(H/M)$ values at $t = 0$ h were distributed quite narrowly around 0 (an experiment/control ratio of 1.0) with a standard deviation of 0.083. Average $\log_2(H/M)$ values for > 99% of all proteins fell within 3 standard deviations, that is within a range of H/M ratios of 0.84–1.18, indicating that changes of > 18% in H/M (experiment/control) ratios were highly unlikely to occur by chance.

In addition to these three experiments, we performed two separate experiments in identical fashion using an alternative proteasomal inhibitor, namely epoxomicin (5 μM; Meng *et al*, 1999). Epoxomicin is an exquisitely selective proteasome inhibitor, which in common with lactacystin, acts by covalently modifying catalytic β-subunits of the proteasome, is even more selective than lactacystin (e.g. does not inhibit Cathepsin A or B; Meng *et al*, 1999; Kozlowski *et al*, 2001), and effectively blocks proteasome function in our preparations (Figs EV1C and EV2). The distribution of average normalized $\log_2(H/M)$ values in these experiments at $t = 0$ was somewhat broader (standard deviation = 0.127; Fig 6C and D), which might be expected given that these were based on two rather than three measurements for each protein.

We then examined how increasingly longer exposures to lactacystin or epoxomicin affected H/M ratios averaged across the entire protein populations. Somewhat surprisingly, the observed changes were very small (Fig 6A and C) and not statistically significant. To increase the statistical power of these analyses, and given the similar mode of action of the two inhibitors, we averaged the normalized $\log_2(H/M)$ values of each protein across all five experiments (3 lactacystin and 2 epoxomicin). Here too, no statistically significant differences were observed (Fig 6E). Although we cannot completely exclude the possibility that the minor differences in experiment/control ratios observed from one time point to another (ranging from −5% to +7%) reflect real effects of proteasome inhibition, we believe that these mainly stem from residual errors in the H/M normalization process described above. Importantly, pooling data across all five experiments increased the resolving power of

our analyses, with the standard deviation of $\log_2(H/M)$ values at $t = 0$ dropping to 0.069 (experiment/control ratio uncertainties of ~14%; Fig 6F). Consequently, data pooled in this fashion were used in all subsequent analyses.

## Effects of proteasomal inhibition on synaptic protein degradation rates

To examine how lactacystin and epoxomicin affect degradation rates of specific proteins, we compiled a stringent list of normalized $\log_2(H/M)$ measurements for all proteins for all five experiments and all four time points (20 samples/protein). Proteins were included in this list if H/M ratio measurements were obtained for at least 2 peptides in each of at least 11 of the 12 lactacystin samples and at least 7 of the 8 epoxomicin samples (176 synaptic proteins, 1,530 proteins in total). We then (i) calculated the mean normalized $\log_2(H/M)$ ratio value for each protein at each time point, and (ii) calculated the statistical significance of the difference between values measured at $t = 24$ h and $t = 0$ h (two-tailed Welch test). The full list is provided in Table EV3.

Of the 1,530 proteins in this list, only 160 (~10%) exhibited statistically significant differences ($P \leq 0.05$) in their normalized $\log_2(H/M)$ ratios at $t = 24$ h. Of these, 158 exhibited positive $\log_2(H/M)$ ratios, that is, exhibited slower degradation rates in the presence of proteasomal inhibitors; of these, 121 (~8% of the 1,530 proteins) exhibited normalized $\log_2(H/M)$ ratios > 3 standard deviations of the $\log_2(H/M)$ ratios measured at $t = 0$ (0.20, i.e. H/M ratios > 1.13; Fig 6F). Gene Ontology analysis (GOrilla) based on the 1,530 proteins of Table EV3 sorted according to $\log_2(H/M)$ ratios at $t = 24$ h indicated that proteins most strongly affected were related to mitochondria ("Component"), to proton transport and to unsaturated fatty acid metabolism ("Process").

Of the 176 synaptic proteins identified here, 21 proteins (~12%) exhibited statistically significant differences (all positive, i.e. slower degradation rates), and for 17 of these, normalized $\log_2(H/M)$ ratios were > 3 standard deviations as described above. The five synaptic proteins that showed the greatest statistically significant slowing of degradation rates were the E3 ligase NEDD4, the atypical A-kinase anchoring protein (AKAP) Neurochondrin/Norbin, the postsynaptic density protein SAP102/Dlg3, the AKAP AKAP9/Yotiao; and the major subunit of AMPA-type glutamate receptors GluA2. Both lactacystin and epoxomicin slowed the degradation of these proteins in similar manners (Fig 7A and B). Given this similarity (see also Appendix Fig S3), we combined the data from both inhibitors in all subsequent analyses, as shown in Fig 7C.

In contrast to the slower degradation observed for these proteins, degradation rates of many other synaptic proteins were not significantly affected. This is shown for three synaptic protein groups in Fig 7D–F and summarized for 45 well-characterized synaptic proteins in Fig 7G. Normalized $\log_2(H/M)$ ratios for the apparently unaffected proteins showed small and uniform changes across time points, but these probably reflect residual errors of the normalization process as mentioned above (compare with Fig 6E). It is worth noting that within the group of apparently unaffected synaptic proteins, a few additional proteins exhibited trends toward change, whose statistical significance was close, but did not quite reach the cutoff of 0.05 (Fig 7G). Setting the cutoff at $P = 0.10$ increased the number of affected proteins to 44 (25%), which still seems to

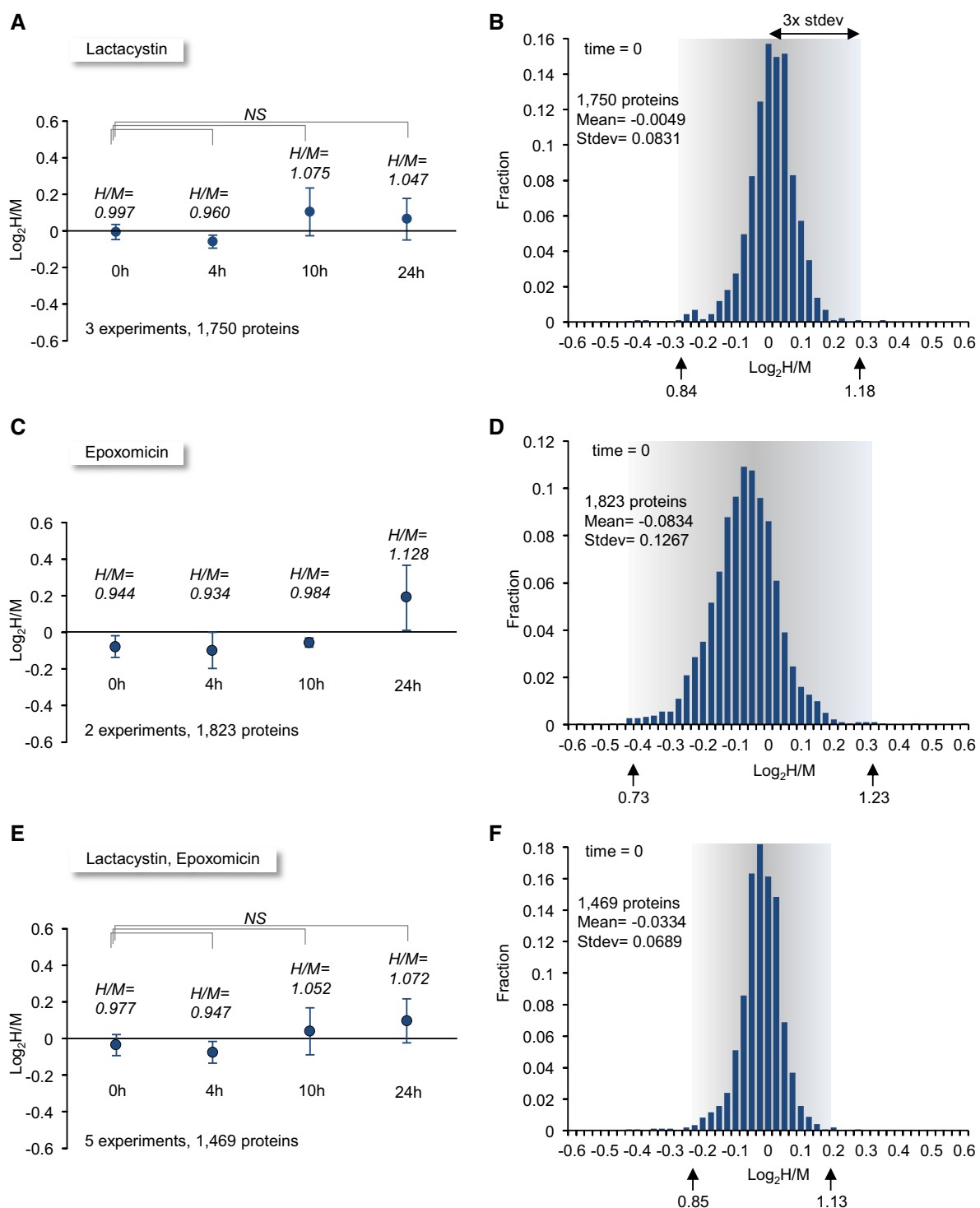

**Figure 6.  Global changes in degradation rates and method sensitivity.**

A    Average log₂(H/M) ratios for 1,750 proteins at four time points in three separate experiments. Each time point shows the average and standard deviation of three mean log₂(H/M) values obtained in three separate experiments. Each mean, in turn, represents the mean log₂(H/M) value measured in the population of 1,750 proteins in that experiment and time point.

B    Distribution of log₂(H/M) values for 1,750 proteins at $t$ = 0. The log₂(H/M) value of each protein represents the average value for that protein in the three experiments. In principle, log₂(H/M) for all proteins at $t$ = 0 should be 0, but in practice they are distributed around 0 as shown here. The H/M ratios for three standard deviation limits are shown beneath the *x*-axis (arrows).

C, D   The same as in (A) and (B) but for two experiments in which epoxomicin was used (1,823 proteins).

E, F   The same as in (A) and (B) but for the combined data from all experiments (3 experiments in which lactacystin was used and 2 experiments in which epoxomicin was used; 1,469 proteins).

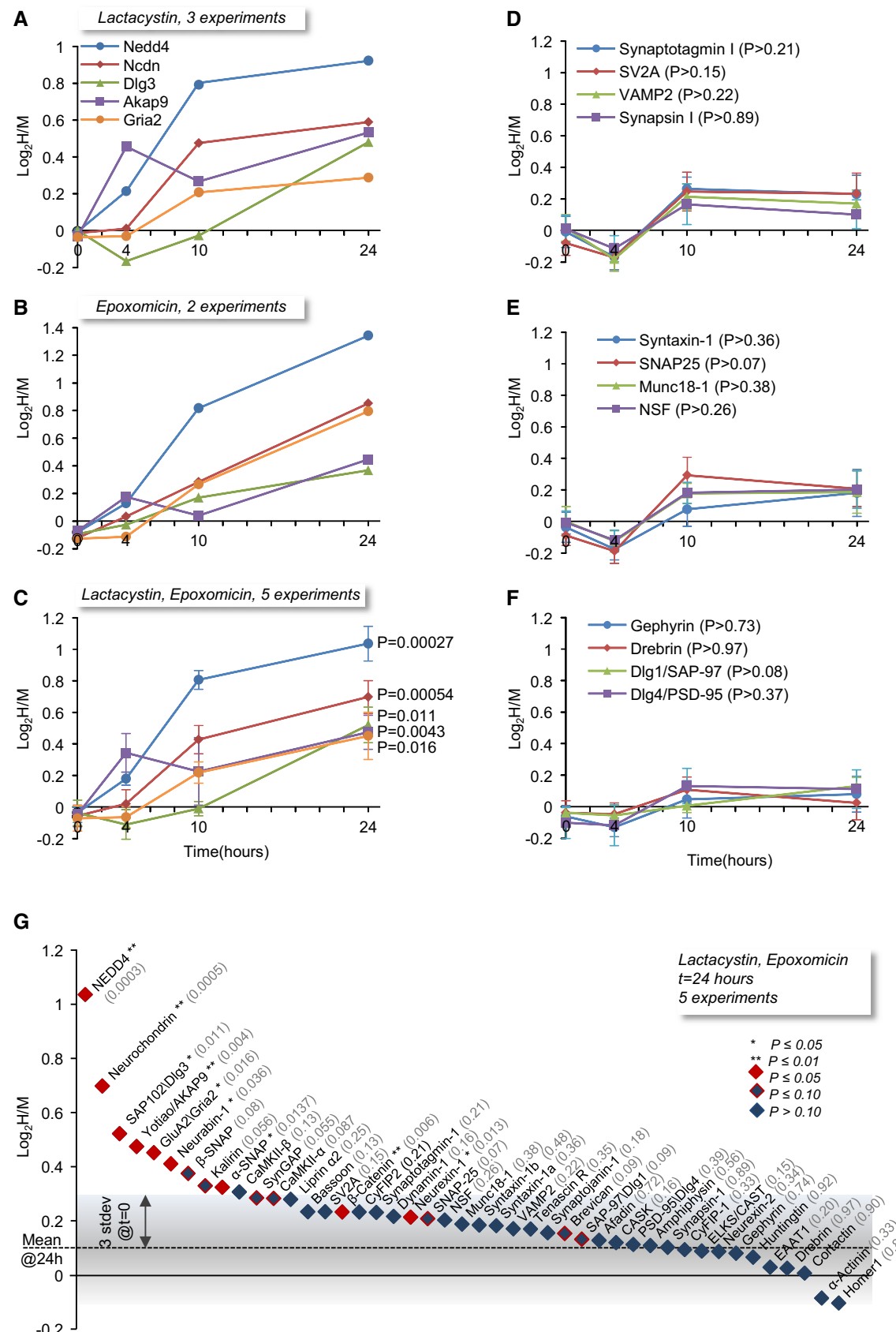

Figure 7.

◀

**Figure 7.   Proteasomal inhibition slows the degradation rates of a subset of synaptic proteins.**

A    $\log_2$(H/M) values for five synaptic proteins for which degradation rates were significantly slowed down in the presence of lactacystin. Each point represents average $\log_2$(H/M) from three experiments.

B    $\log_2$(H/M) values for the same five synaptic proteins in experiments in which epoxomicin was used. Each point represents average $\log_2$(H/M) from two experiments.

C    Average ($\pm$ SEM) $\log_2$(H/M) ratios for the same five synaptic proteins, based on all five experiments (3 lactacystin, 2 epoxomicin). *P*-values were calculated using the two-tailed Welch's *t*-test, comparing the five $\log_2$(H/M) measurements at *t* = 24 to those measured for the same proteins at *t* = 0.

D–F  Average ($\pm$ SEM) $\log_2$(H/M) values for three groups of synaptic proteins (synaptic vesicle, presynaptic, postsynaptic), based on all five experiments. None of the changes observed for these proteins were statistically significant (two-tailed Welch's *t*-test). The temporal changes common to all proteins and all groups most likely represent slight imperfections in the H/M normalization process (compare with Fig 6E; see the Results section for further details).

G    $\log_2$(H/M) values at *t* = 24 h for a group of well-studied synaptic proteins, sorted from highest to lowest. *P*-values (two-tailed Welch's *t*-test) of comparisons with $\log_2$(H/M) values for the same proteins at *t* = 0 are shown in parentheses. Proteins that exhibited statistically significant changes ($P \le 0.05$) at *t* = 24 h are shown in red. Proteins for which ($0.05 < P \le 0.10$) at *t* = 24 h are shown in blue/red. The gray region depicts three standard deviations of $\log_2$(H/M) values at *t* = 0 (Fig 6F), to provide a measure of the method's sensitivity.

suggest that under our experimental conditions, lactacystin and epoxomicin did not significantly slow the degradation rates of the majority of synaptic proteins.

**Relations of proteasomal inhibition with protein half-lives**

Examination of the list of synaptic proteins whose degradation was most strongly affected by lactacystin and epoxomicin reveals an interesting trend: Practically, all of these—NEDD4, Neurochondrin, SAP102/Dlg3, GluA2, Neurabin-1—were previously shown to be particularly short-lived when compared to the typical half-lives of synaptic proteins (Cohen *et al*, 2013). Specifically, the half-lives of these proteins ranged from 0.72 for NEDD4 to 2.13 days for SAP102/Dlg3, much shorter than the average half-life of synaptic proteins in these preparations (4.14 days; Cohen *et al*, 2013). The same holds for AKAP9/Yotiao, whose half-life was estimated at 1.31 days in our current experiments (Appendix Fig S2; Table EV2). While this observation is consistent with the notion that UPS-mediated degradation is a favored route for short-lived proteins, it is important to keep in mind that in any experiment (including our own) in which degradation pathways are inhibited, the strongest effects will be observed for proteins with the shortest half-lives. In fact, for a protein whose degradation occurs mainly through the UPS, simple assumptions allow one to predict expected H/M ratios for this protein at different time points from its known half-life ($t_{1/2}$) and the degree of proteasomal inhibition (Fig 8A). Specifically, assuming that for this protein (i) degradation occurs as a first order reaction such that its residual amount decays with a time constant of $\tau$ (with $\tau \approx 1.443 \cdot t_{1/2}$), and that (ii) slowing of its degradation by proteasomal inhibition increases $\tau$ by some factor $\alpha$, then value of $\log_2$(H/M) for this protein at time *t* is expected to be:

$$\log_2\left(\frac{\text{H}}{\text{M}}\right) = \frac{t(\alpha - 1)}{\alpha t_{1/2}} \tag{1}$$

(see Appendix Supplementary Materials and Methods for a detailed derivation).

As shown in Fig 8B, for *t* = 24 h expected $\log_2$(H/M) values drop rapidly with increasing protein longevity. Conversely, dependence on $\alpha$ saturates very quickly, such that the effects of 10×, 50×, or 100× inhibition of UPS-based degradation are practically identical. This is expected, because at large values of $\alpha$, equation 1 is effectively reduced to:

$$\log_2\left(\frac{\text{H}}{\text{M}}\right) \approx \frac{t}{t_{1/2}} \quad (\text{for } \alpha \gg 1) \tag{2}$$

Given that the concentrations of lactacystin and epoxomicin used here reduced proteasomal function at least 10-fold (Figs 1, EV1 and EV2) and the insensitivity to exact values of $\alpha$, a value of $\alpha = 10$ was used hereafter. Equation 1 also suggests that profiles such as those shown in Fig 7A–C are also very sensitive to the half-life of the protein in question (Fig 8C). In fact, this figure suggests that for particularly long-lived proteins, the ability to resolve the effects of pharmacological inhibitors on degradation rates becomes questionable. We will return to this point later.

The analytical approach described above was applied to the experimental data. To that end, we plotted, for each protein, the average $\log_2$(H/M) value obtained at 0, 4, 10, and 24 h as a function of its half-life. Half-life estimates for 1,177 proteins were obtained from Cohen *et al* (2013). Estimates for an additional set of 239 proteins were obtained from the current data set (Appendix Fig S2; Table EV2), resulting in half-life estimates for 1,416 proteins in total. As shown in Fig 9, proteins for which statistically significant changes were observed at 24 h, tended to distribute along expected $\log_2$(H/M) value curves generated according to equation 1. This tendency was most obvious for short-lived proteins, but not exclusive to this group. Conversely, proteins for which statistically significant changes were not observed did not tend to distribute along the predicted curves; this was particularly true for long-lived proteins but was also observed for numerous short-lived ones. The different tendencies observed qualitatively were confirmed quantitatively by comparing the correlation (Pearson's) between measured and expected $\log_2$(H/M) values for proteins with $t_{1/2} \le 5$ days belonging to the two groups. These were $r = 0.75$, for proteins with statistically significant changes at 24 h (i.e. $P \le 0.05$; 142 proteins), $r = 0.15$, for proteins without statistically significant changes (640 proteins) and $r = 0.01$ for the latter group after removing one extreme outlier (Kif1a; 639 proteins).

A similar analysis was carried out for 174 synaptic proteins for which half-life estimates were available (Fig 10). Here too, the trends were similar: Proteins for which statistically significant changes were observed at 24 h distributed along the curves generated according to equation 1. Conversely, proteins for which statistically significant changes were not observed (including short-lived ones) were more broadly distributed, and did not relate to the predicted values.

The fit between expected and measured values for proteins for which statistically significant changes in H/M ratios were detected strongly reinforces the conclusion that their degradation is indeed sensitive to UPS inhibition. Similarly, the poor fit for the rest of the proteins is in line with the failure to detect statistically significant

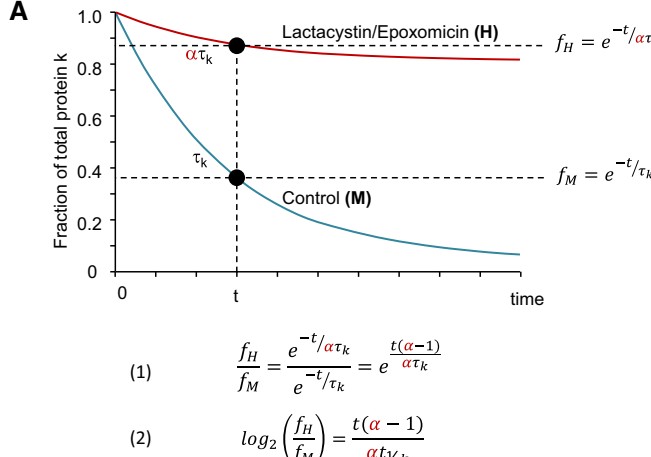

$$(1) \qquad \frac{f_H}{f_M} = \frac{e^{-t/\alpha\tau_k}}{e^{-t/\tau_k}} = e^{\frac{t(\alpha-1)}{\alpha\tau_k}}$$

$$(2) \qquad log_2\left(\frac{f_H}{f_M}\right) = \frac{t(\alpha-1)}{\alpha t_{1/2 k}}$$

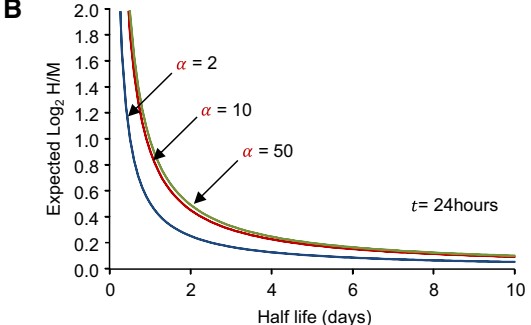

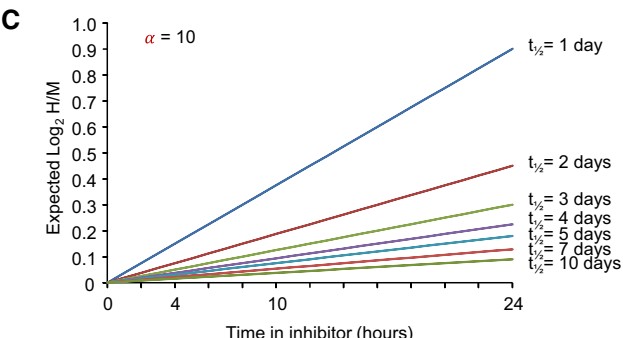

**Figure 8. Expected dependence of log₂(H/M) on protein half-life.**

A  An illustration of the quantitative framework constructed to relate expected log₂(H/M) measurements to the known half-life of a particular protein. For the sake of simplicity, the natural logarithm (*e*) is used here, but values were ultimately converted to base 2 (which is the base used in the half-life terminology).

B  Dependence of expected log₂(H/M) values at *t* = 24 h on the protein's half-life, given the factor (α) by which its catabolism is inhibited. Note that beyond α = 10, the curves are nearly identical.

C  Expected log₂(H/M) values over the course of 24 h for proteins with half-lives ranging from 1 to 10 days. Compare with the actual measurements shown in Fig 7A–C.

effects of proteasomal inhibitors on their degradation. The conclusion that the degradation of particular proteins is insensitive to UPS inhibition has, however, an important caveat: Where proteins with long half-lives are concerned, our analysis indicates that the

expected log₂(H/M) values, even at the latest time point, should be so small that they might elude detection (Fig 9D). Thus, for example, for proteins with half-lives of 5 days or longer, the expected log₂(H/M) value at the 24-h point is less than ~0.18 (H/M ≈ 1.13) which is near the limit of our H/M ratio measurement certainty (Fig 6E). Therefore, we cannot exclude the possibility that the degradation of some long-lived proteins ($t_{1/2} > 5$ days) is mediated directly or indirectly by the UPS, even though statistically significant changes were not detected. It is important to note in this regard that this conclusion is not limited to our experiments, as it applies equally to any experimental approach based on protein quantification following exposure to protein degradation inhibitors. This matter is addressed further in the Discussion.

### Effects of proteasomal inhibitors on synaptic protein synthesis

In the first set of SILAC experiments described above (Figs 3 and 4), we found that lactacystin strongly affected the newly synthesized/preexisting (H/L) ratios measured for synaptic proteins, but for reasons explained above, these experiments could not discern if these effects reflected reduced degradation of preexisting proteins or reduced synthesis of new ones. The second set of experiments (Figs 6–10), however, provided selective information on protein degradation. By combining information from the two data sets, the effects of proteasomal inhibitors on synaptic protein synthesis could now be disambiguated. To that end, we compiled a list of synaptic proteins for which the first set of experiments suggested apparent half-live prolongation in the presence of lactacystin (H/L ratios significantly reduced by more than three standard deviations of the variability observed in control experiments; gray region in Fig 4A) and then used the second data set (Figs 6–10; Table EV3) to determine whether the degradation of these proteins was significantly inhibited. Of the 79 synaptic proteins in this list, the degradation of only 11 (< 14%) was affected in a statistically significant manner, indicating that changes in H/L ratios observed in the first set of experiments were caused primarily by substantial inhibition of protein synthesis (manifested as reduced incorporation of heavy AAs). It is worth pointing out that the scarcity of heavy AA-labeled peptides in lactacystin-treated cells (< 2 peptides in either of the two experiments) precluded the inclusion of 33 additional synaptic proteins in the aforementioned list (including Syntaxin-1A, RIM1, GluN2B, GRIP1, ProSAP1/Shank2, GABA_B1, GABA_B2, Neuroligin3) even though heavy peptides for these proteins were in present in sufficient quantities in untreated cells (> 2 peptides in each of both experiments). These findings thus suggest that one of the strongest effects proteasomal inhibitors have on synaptic protein metabolism is a dramatic suppression of their synthesis.

Careful examination of the identities of proteins for which considerable elevations in H/L ratios were observed in the presence of lactacystin (Fig 4A, right hand side of plot; see list in Table EV1) provides clues as to pathways involved in the suppression of (synaptic) protein synthesis. This list contains many chaperones (including the critical ER chaperone, GRP78/Hspa5), 28 proteasome subunits (8 of which exhibit positive elevations > 3 standard deviations) as well as ubiquitin, suggesting *accelerated synthesis* of these proteins in response to UPS inhibition. These data are in line with prior studies suggesting that UPS inhibition

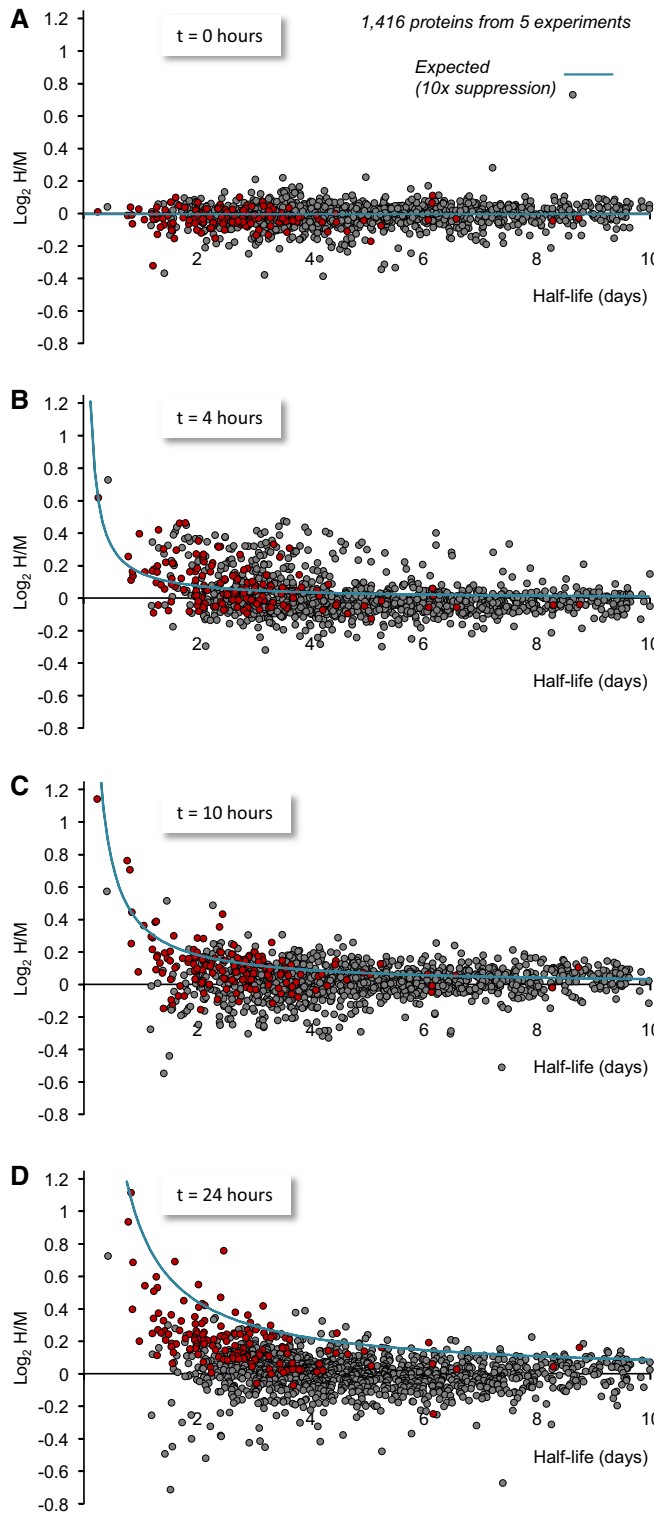

**Figure 9.  Dependence of log₂(H/M) values on protein half-lives.**

A–D   Measured $\log_2$(H/M) values at four time points as a function of their half-lives estimated here and elsewhere (Cohen *et al*, 2013). All $\log_2$(H/M) values shown here are averages of values obtained in five experiments (3 using lactacystin, 2 using epoxomicin). Proteins for which statistically significant differences at $t = 24$ h were observed ($P < 0.05$, two-tailed Welch's *t*-test) are shown in red. Expected $\log_2$(H/M) values based on equation 1 are plotted as light blue lines. To minimize the masking of potential dependencies by the slight imperfections in the H/M normalization process (which introduces small offsets along the *y*-axis), average population $\log_2$(H/M) values of each time point (Fig 6E) were subtracted from all measured $\log_2$(H/M) values. Data for 1,416 proteins for which half-life estimates were available.

the enhanced synthesis of molecules critically involved in handling excess misfolded proteins, including ER-localized chaperones and folding enzymes (reviewed in Walter & Ron, 2011). The multiple chaperones with elevated H/L ratios in the presence of lactacystin (indicative of accelerated synthesis) as well as the widespread suppression of protein synthesis are in good agreement with this interpretation.

Under normal circumstances, unfolded/misfolded ER proteins are degraded by the proteasome as the last step of a process known as ER-associated degradation (ERAD; Brodsky, 2012; Olzmann *et al*, 2013; Ruggiano *et al*, 2014). Similarly, the proteasome plays key roles in the degradation of defective ribosomal products (DRiPs), newly synthesized polypeptides that, for a variety of reasons, fail to fold properly. Interestingly, at the earliest time point (4 h) of the experiments of Figs 5–10, inhibition of degradation was noted for a conspicuous group of proteins, which seemed to occur irrespective of their half-lives (Fig 9B). We examined the proteins for which, at $t = 4$ h, $\log_2$(H/M) values were in the top 5%, that is, proteins whose degradation was most strongly suppressed at this time (70 proteins) and noted that indeed, correlation with $\log_2$(H/M) values expected from their known half-lives (equation 1) was rather poor ($r = 0.18$, excluding two extreme outliers, Kif1a and Scd2). It should be noted that for many of these proteins, the statistical significance of changes in $\log_2$(H/M) values at $t = 4$ h as compared to $t = 0$ h was not high ($P < 0.10$: 26/70 proteins; $P < 0.05$: 10/70 proteins), but as a group, they seemed to stand out. We suspect that this group reflects, for the most part, effects of UPS inhibition on the degradation of recently synthesized proteins (i.e. synthesized shortly before inhibitors were added) by quality control mechanisms (e.g. ERAD, DRiP), rather than functional, short-lived pools of these proteins. This interpretation is based on the following reasoning: First, we noted that most proteins in this group (top 5%) were particularly large, with an average molecular weight of 271 kDa (in fact, the molecular weight of only 2 proteins was < 110 kDa, one of these being ubiquitin). This is nearly four times greater than the average molecular weight of proteins in our stringent list (71 kDa; Table EV3). It might be reasonable to expect that large proteins are particularly prone to folding errors and subsequently to more degradation via quality control-related mechanisms (Duttler *et al*, 2013). Second, MS-based quantifications provide bulk information on peptides and proteins, and thus for a particular pool to affect the MS read-out, it would have to constitute a substantial fraction of this bulk. Indeed, early studies have claimed that DRiPs constitute upwards of 30% of newly synthesized proteins (Schubert *et al*, 2000) and

leads to a generalized cellular response known as the unfolded protein response (UPR) (Obeng *et al*, 2006; Larance *et al*, 2013), a cellular reaction triggered by the buildup of unfolded/misfolded proteins in the endoplasmic reticulum (ER). This cellular response includes, among others, a generalized shutdown of protein synthesis (reviewed in Chakrabarti *et al*, 2011; Hetz *et al*, 2015; see also Shalgi *et al*, 2013; Halliday & Mallucci, 2014) and, paradoxically,

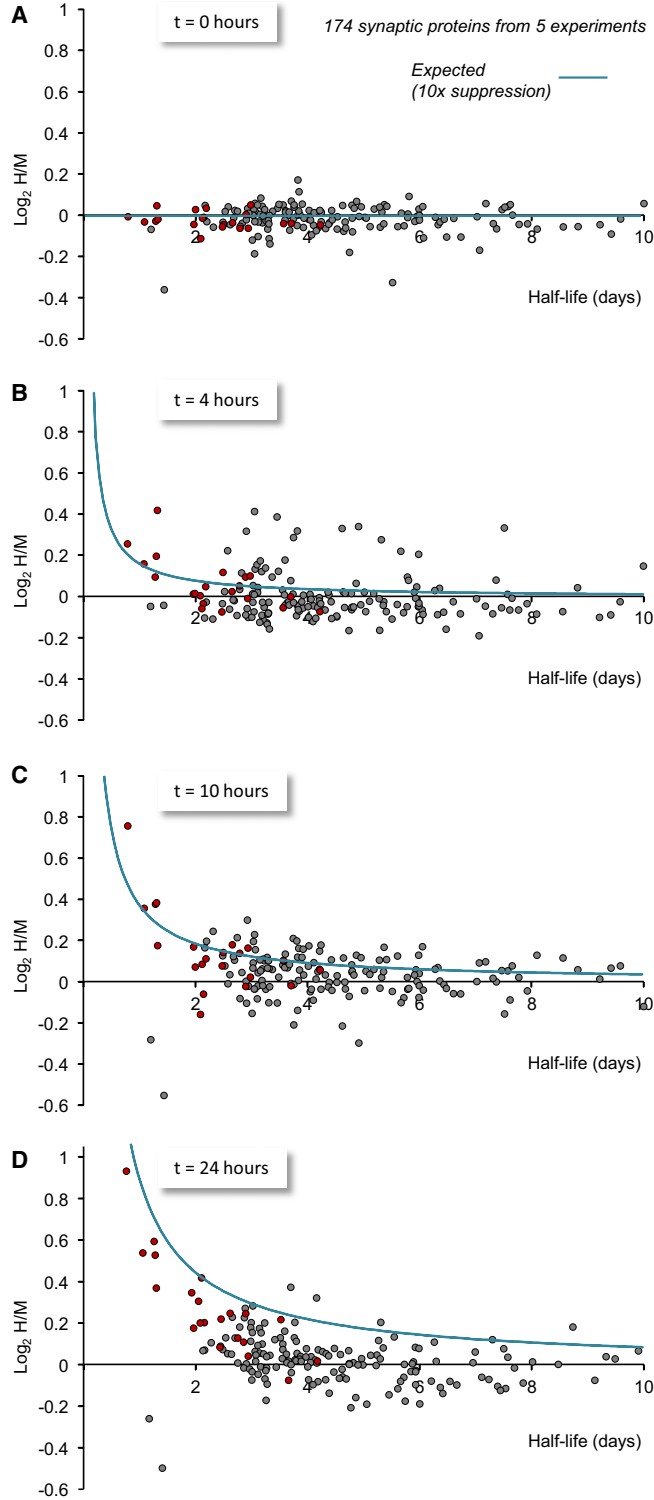

**Figure 10. Dependence of log$_2$(H/M) values on synaptic protein half-lives.**

A–D Measured log$_2$(H/M) values at four time points as a function of their half-lives estimated here and elsewhere (Cohen *et al*, 2013). All log$_2$(H/M) values shown here are averages of values obtained in five experiments (3 using lactacystin, 2 using epoxomicin). Proteins for which statistically significant differences at *t* = 24 h were observed (*P* < 0.05, two-tailed Welch's *t*-test) are shown in red. Expected log$_2$(H/M) values based on equation 1 are plotted as light blue lines. To minimize the masking of potential dependencies by the slight imperfections in the H/M normalization process (which introduces small offsets along the *y*-axis), average population log$_2$(H/M) values of each time point (Fig 6E) were subtracted from all measured log$_2$(H/M) values. Data for 174 synaptic proteins for which half-life estimates were available.

dependence, the sizes of the affected proteins and their substantial pools would seem to suggest that our interpretation is the most parsimonious.

## Discussion

In the current study, we used pulsed and multiplexed SILAC combined with MS to systematically and selectively measure how pharmacological suppression of proteasomal activity affects degradation (and synthesis) rates of thousands of neuronal and synaptic proteins. Using these approaches, we identified a group of synaptic (and non-synaptic) proteins whose degradation appeared to be strongly dependent on UPS function. We also found, however, that the degradation of most synaptic (and non-synaptic) proteins was not significantly slowed down in the presence of proteasomal inhibitors. By combining these data with information on metabolic turnover rates of individual proteins, we devised a simple quantitative framework that explained some of the differential effects of proteasomal inhibition on protein degradation. This framework, however, also delineated important limitations in the use of catabolic inhibitors to study degradation of long-lived proteins. In contrast to the restricted effects on protein degradation, UPS inhibition profoundly affected the synthesis of numerous proteins, probably by inducing an unfolded protein response, among others. Synaptic proteins were particularly prominent within the group of proteins whose synthesis was suppressed, indicating that synaptic protein synthesis is exquisitely sensitive to this cellular response. Our findings thus point to specific proteins whose degradation is UPS mediated, but also indicate that under basal conditions, most synaptic proteins are degraded through alternative pathways.

### Methodological considerations

A common method for studying the involvement of catabolic pathways in the degradation of particular proteins is to pharmacologically inhibit the pathways, and examine—by Western blots or quantitative immunohistochemistry, for example—how these manipulations affect total quantities of the proteins in question. Unfortunately, these methods are incapable of separating the effects of such inhibitors on protein degradation from effects on protein synthesis. These processes are more readily discernible in pulse-chase experiments based on radioactive amino acids; this approach, however, is not free from confounds either: On the one hand, the

more recent studies estimate that as much as ~15% of nascent peptide chains whose properties preclude efficient folding are cotranslationally tagged for degradation (Duttler *et al*, 2013; Wang *et al*, 2013; reviewed by Pechmann *et al*, 2013). Thus, while we cannot fully exclude the existence of functional protein pools with half-lives shorter than prior estimates, the short timescales, UPS

slow turnover rates of synaptic proteins require relatively long labeling periods. On the other, labeling is typically associated with substantial (ten- to hundredfold) reductions in (labeled) amino acid concentrations, raising the risk of inducing macroautophagy (see, e.g. Sutter *et al*, 2013). In contrast, the methods used here do not suffer from either of these shortcomings. Moreover, their sensitivity is quite high, exhibiting an ability to detect differences as small as 14% with high certainty (Fig 6).

Although the approach taken here has many advantages, it is not free from potential confounds. The first concerns the interpretation of findings based on pharmacological inhibitors. Thus, for example, slower degradation of a particular protein in the presence of proteasomal inhibitors cannot be taken as unequivocal evidence for direct degradation by the UPS; the slowing down could reflect, for example, changes in upstream molecules that regulate its degradation (as discussed below for GluA2). Conversely, the observation that its degradation is not slowed might result from the rerouting of its degradation through alternative catabolic pathways which are known to be intricately interconnected. Thus, for instance, UPS inhibition by MG-132 or lactacystin has been shown to activate macroautophagy and upregulate levels of lysosomal enzymes (e.g. Ding *et al*, 2007b; Shen *et al*, 2013). Indeed, we noted that fits to expected $\log_2(H/M)$ values were somewhat better at 10 h as compared to fits at 24 h (Figs 9 and 10), suggesting that at the later time point, degradation rates were not slowed down to the same extent, possibly implying that after an initial delay, degradation of some proteins commenced through alternative routes.

A second confound concerns the ability to evaluate the roles of UPS-based degradation on the catabolism of long-lived proteins. Simple considerations, exemplified in Figs 8–10, indicate that for particularly long-lived proteins, the expected extent of protein loss over short time periods is so small that it should elude detection by all but the most sensitive assays. Along this line, we concluded that our assay, sensitive as it might be, was probably insufficiently sensitive to detect changes in the abundance of proteins whose half-lives are > 5 days even when proteasomal inhibitors were applied for 24 h. We thus find it puzzling that prior studies have reported significant accumulation of synaptic proteins with known half-lives of 3 to 5 days, after inhibiting proteasomal function for 4–8 h or even less. It could be argued that bulk degradation rates measured here are slower than degradation rates of the same proteins at synapses. We note, however, that in our hands, little if any accumulation of some of the very same proteins was observed at synaptic sites after 10-h exposures to lactacystin (Fig 2; see also Shin *et al*, 2012). Moreover, it is hard to reconcile this argument with the fact that many of the synaptic proteins studied here are strongly localized to synaptic sites, and thus synaptic pools represent the bulk of these proteins in neurons. We note, however, that in many of these studies, effects of proteasomal inhibition were examined following acute manipulations of neuronal activity (Ehlers, 2003; Willeumier *et al*, 2006; Guo & Wang, 2007; Saliba *et al*, 2007; Jiang *et al*, 2010; Lazarevic *et al*, 2011; Shin *et al*, 2012), and therefore, the rapid effects of proteasomal inhibitors might reflect transients or adjustments associated with strong changes in activity levels. The current study, on the other hand, was focused on basal degradation processes in the absence of overt manipulations of network activity.

Having said this, we also noted that the degradation of several proteins was substantially retarded at the earliest time point (4 h;

Fig 9B) in manners unexpected from their known half-lives. As mentioned above, we hypothesize that this reflects the effects of UPS inhibition on the degradation of recently synthesized proteins targeted to the UPS as part of cellular quality control mechanisms rather than the degradation of short-lived, functional pools of these proteins. In fact, it has been suggested that this early processing stage represents a major station for regulating the amounts of proteins ultimately trafficked to synapses (e.g. following changes in activity levels; Saliba *et al*, 2007). A related suggestion is the "stabilized binding complex" model (Cambridge *et al*, 2011) which posits that components of multimolecular complexes (as synaptic proteins tend to be) are produced in excess such that protein copies not integrated into their cognate complexes are rapidly degraded, resulting in a simple mechanism for producing components at correct stoichiometries.

The conspicuous effects of proteasomal inhibitors at the 4-h time point but not later are in agreement with prior reports. For example, 10–40% increases in cellular levels of several synaptic proteins were observed after 6 h but not after 24 h in the presence of MG-132 (Lazarevic *et al*, 2011). Interestingly, genes encoding for proteins expressed in response to ER stress were found to be upregulated ~4 h after lactacystin treatment of cultured murine cortical neurons (Choy *et al*, 2011). Furthermore, among the proteins whose degradation was retarded at this early time point we found BCL2-associated athanogene 6 (BAG6), a protein heavily involved in handling misfolded substrates (including DRiPs) and protecting them from aggregation on their way to the proteasome (Minami *et al*, 2010; Xu *et al*, 2013). Perhaps the rapid cellular response, combined with the suppression of protein synthesis (see below), limits the obvious manifestation of this process to the first few hours of proteasomal inhibition.

Finally, when we examined spontaneous network activity levels following exposure to lactacystin, no particular trends were observed over a period of about one day (Appendix Fig S4). As spontaneous network activity in these preparations strongly depends on synaptic transmission (Eytan & Marom, 2006), this further suggests that synaptic function was not grossly impaired by suppressing proteasomal activity for 10-20 h. This should not be taken to imply, however, that the long-term effects of proteasomal inhibition were negligible: Morphological measures of neurons and synapses (counts, sizes), not noticeably altered after 24 h (Fig EV5A,B,D and E; see also Figs 1B and 2C), showed unmistakable signs of deterioration after 48 h (Fig EV5C–E), in full agreement with prior studies (e.g. Dantuma *et al*, 2000). This also precluded extension of our proteomic analysis to durations beyond 24 h.

## Proteasome inhibition and synaptic protein degradation

Our unbiased approach identified a group of proteins whose degradation was significantly slowed down in the presence of selective proteasomal inhibitors. The majority of these have been previously shown to be relatively short-lived (Appendix Fig S2; Table EV2; Cohen *et al*, 2013), in line with the generally accepted notion that the proteasome preferentially degrades short-lived proteins. As we show, however, the prevalence of short-lived proteins within this group reflects, in part, the unavoidable fact that blocking a catabolic process will necessarily have the largest effects on the shortest-lived species (see Fig 8). Nevertheless, the degradation of many

short-lived proteins was not significantly slowed by proteasomal inhibitors (Figs 9 and 10), indicating that UPS-mediated degradation was limited to a population of specific proteins.

Among the group of proteins whose degradation was significantly retarded we found multiple proteins involved in glutamate receptor trafficking, including the AMPA-type glutamate receptor (AMPAR) subunit GluA2. This is interesting because regulation of AMPAR content is widely believed to be a central control point for regulating synapse strength (Huganir & Nicoll, 2013). All four AMPAR subunit types (GluA1–4) can undergo activity-dependent ubiquitination, and it is thought that AMPAR subunit ubiquitination determines whether internalized AMPARs are targeted for lysosomal degradation or allowed to recycle back to the membrane (reviewed in Goo *et al*, 2015). Interestingly, one of the proteins whose degradation was most strongly affected here was the E3 ligase NEDD4, which has been shown in multiple studies to ubiquitinate AMPARs and regulate their targeting for degradation (Schwarz *et al*, 2010; Hou *et al*, 2011; Lin *et al*, 2011; Lin & Man, 2014; Scudder *et al*, 2014) in a manner that is counteracted by the DUBs Usp8 and Usp46 (Scudder *et al*, 2014; Huo *et al*, 2015). It has also been suggested that AMPA receptors might be degraded directly by the proteasome (Hou *et al*, 2011; see also Zhang *et al*, 2009; Fu *et al*, 2011), but this conclusion was based to a large degree on the use of proteasomal inhibitors, which, as we show here, also affect the degradation of the upstream regulator NEDD4. It thus remains possible that the slowing of GluA2 degradation by proteasomal inhibitors reported here (Fig 7) represents an indirect effect mediated by upstream regulatory molecules as discussed in the prior section.

Several other proteins within the group of affected proteins were previously shown to be involved in glutamate receptor trafficking: Neurochondrin/Norbin is thought to modulate the trafficking and function of metabotropic glutamate receptor 5 (Wang *et al*, 2009; Hermann *et al*, 2015); SAP102/Dlg3 is a postsynaptic density protein involved in glutamate receptor targeting, trafficking, and clearance (Zheng *et al*, 2011; Chen *et al*, 2012); interestingly, SAP102/Dlg3 trafficking was recently shown to be regulated by NEDD4-mediated monoubiquitination during apical tight junction formation (Van Campenhout *et al*, 2011); AKAP9/Yotiao was originally suggested to regulate glutamate receptor activity (Westphal *et al*, 1999); Neurabin—a synaptic actin binding protein—has been shown to target protein phosphatase (PP1) to the synapse to regulate distinct AMPAR trafficking pathways (Hu *et al*, 2007); finally, SNAP-α was shown to interact directly with GluA2 and was implicated in AMPAR trafficking (Osten *et al*, 1998; Hanley *et al*, 2002; Hanley, 2007). It should be noted that the molecules listed above are involved in many other cellular processes; yet the association with glutamate receptor trafficking is nonetheless intriguing.

The absence of several protein groups from the list of proteins whose degradation was significantly slowed down is quite conspicuous. Thus, for example, very few synaptic vesicle molecules or molecules involved in SV exocytosis or endocytosis were present in this list. Furthermore, degradation of numerous proteins linked to neurodegenerative diseases such as Huntingtin (htt; $t_{1/2} \approx 3.7$ days) α-synuclein (Snca; $t_{1/2} \approx 2.4$ days), Tau (Mapt; $t_{1/2} > 20$ days), and TDP-43 (Tardbp; $t_{1/2} \approx 5.4$ days) was not significantly retarded by proteasomal inhibitors. For the latter two, and in particular Tau, their long half-lives might have precluded our ability to detect

effects on degradation rates (see above). For the first two, however, we should have been able to detect a change. These data thus suggest that under basal conditions, proteins belonging to these groups might be degraded by pathways that do not depend on proteasomal function.

### Proteasome inhibition and synaptic protein synthesis

In contrast to the rather modest effects of proteasome inhibitors on protein degradation, the effects on protein synthesis were profound. This finding is in good agreement with several prior studies in which the effects of proteasome inhibitors on protein synthesis were examined (Ding *et al*, 2006; King *et al*, 2008; Larance *et al*, 2013; Milner *et al*, 2013). As described in the Results, the proteomic profile was suggestive of an unfolded protein response (Bush *et al*, 1997; Sitia & Braakman, 2003; Zhang & Kaufman, 2004) and possibly of the closely related integrated stress response (as evidenced by increased expression of Clic4, Hmox1, Lonp1, Lgals3, and EPRS; Harding *et al*, 2003; Young & Wek, 2016). The possible induction of an UPR by proteasomal inhibitors has been documented in several previous reports (e.g. Obeng *et al*, 2006; Zhang *et al*, 2010; Larance *et al*, 2013) and is exploited in the treatment of certain malignant diseases by administering proteasome inhibitors approved for clinical use (Clarke *et al*, 2014). More recently, it has been proposed that an UPR occurs in a host of neurodegenerative diseases associated with protein aggregation (Alzheimer's disease, Parkinson's disease, and Prion disease) and that the reactive suppression of protein synthesis catastrophically accelerates synaptic loss and exacerbates the damage caused by the initial precipitating factors (reviewed in Halliday & Mallucci, 2014, 2015; Freeman & Mallucci, 2016; see also Cornejo & Hetz, 2013). Our analysis, covering > 1,400 proteins (Fig 4 and Table EV1), suggests that an UPR induced by proteasomal inhibitors can suppress the synthesis of synaptic proteins by factors of eight or more, and is thus in full agreement with this suggestion. Moreover, the fact that synaptic proteins were the most strongly affected group (Figs 4 and EV3) extends this suggestion and indicates that synapses might be exquisitely sensitive to pathological processes that negatively impact protein synthesis.

In summary, the approach used here allowed us to identify synaptic proteins whose degradation seems to be mediated directly or indirectly by the ubiquitin-proteasome system, and tentatively conclude that many, if not most synaptic proteins seem to be degraded under basal conditions through pathways that do not depend on proteasomal function. Our findings also highlight, however, important limitations in the use of proteasomal inhibitors to study synaptic protein catabolism, which mainly stem from the relatively slow turnover rates of synaptic proteins and the strong effects proteasomal inhibitors have on their synthesis.

# Materials and Methods

### Animal welfare

All experiments were performed in primary cultures of newborn rat neurons prepared according to a protocol approved by the "Technion, Israel Institute of Technology Committee for the Supervision of Animal Experiments" (IL-019-01-13).

## Proteasomal inhibitors

Lactacystin (Santa Cruz Biotechnology) and epoxomicin (Sigma) were stored as concentrated stocks in $H_2O$ and DMSO, respectively, at −20°C. As a precaution, every inhibitor stock used throughout this study was tested for activity by probing Western blots of neuronal cell culture extracts treated with the inhibitor for increasing durations as described below (Western blots). One example is shown in Appendix Fig S5.

## Cell culture

Primary cultures of rat cortical neurons used for SILAC experiments were prepared as described previously (Minerbi *et al*, 2009). Briefly, cortices of 1- to 2-day-old Wistar rats of either sex were dissected, dissociated by trypsin treatment followed by trituration using a siliconized Pasteur pipette. $1.2 \times 10^6$ cells were then plated in 12-well plates whose surface had been pretreated with polyethylenimine (Sigma) to facilitate cell adherence. Cells were initially grown in medium containing minimal essential medium (MEM; Sigma), 25 mg/l insulin (Sigma), 20 mM glucose (Sigma), 2 mM L-glutamine (Sigma), 11.16 mg/l gentamicin sulfate (Sigma), 10% NuSerum (Becton Dickinson Labware), and 0.5% fetal bovine serum (HyClone). The preparation was then transferred to a humidified tissue culture incubator and maintained at 37°C in a 95% air and 5% $CO_2$ mixture. Half the volume of the culture medium was replaced three times a week with feeding medium similar to the medium described above but devoid of NuSerum, containing a lower concentration of L-glutamine (Sigma, 0.5 mM), and 2% B-27 supplement (Gibco).

Primary cultures of rat cortical neurons for live imaging and electrophysiology recording experiments were prepared as described above except that here the neurons were plated onto 22 × 22 mm coverslips inside 6- or 8-mm-diameter glass cylinder (Bellco Glass) microwells, on glass-bottomed 35-mm petri dishes (World Precision Instruments) or on MEA dishes as in Minerbi *et al* (2009).

Primary cultures of rat hippocampal neurons experiments were prepared as described previously (Tsuriel *et al*, 2006). In brief, hippocampal CA1-CA3 regions of 1- to 2-day-old Wistar rats of either sex were dissected and dissociated as described above and plated onto 22 × 22 mm coverslips coated with poly-D-lysine (Sigma) inside 8-mm-diameter glass cylinder microwells. Culture medium consisted of MEM (Gibco), 20 mM glucose, 0.1 g/l bovine transferrin (Calbiochem), 25 mg/l insulin (Sigma), 2 mM L-glutamine (Sigma), 10% NuSerum (Becton Dickinson Labware), 0.5% fetal bovine serum (HyClone), 2% B-27 supplement (Gibco), and 8 μM cytosine β-D-arabinofuranoside (Sigma) which was added to the culture medium after 3 days. Cultures were maintained at 37°C in a 95% air and 5% $CO_2$ humidified incubator. Culture medium was replaced every 7 days.

## Fluorogenic substrate assays *in vitro* and in living cells

Cortical neurons grown for 2 weeks in culture were treated with lactacystin (10 μM), epoxomicin (5 μM) or carrier solution for 4 h, washed vigorously with phosphate-buffered saline (PBS) after which they were scraped in PBS. The lysed cells were centrifuged for 5 min at 600 × *g*. Lysis solution containing Tris-Cl (50 mM), DTT (1 μM), NaCl (150 mM), $MgCl_2$ (1.5 mM), EDTA (1 mM), PMSF (1 mM), ATP (2 mM), iodoacetamide (5 mM), NaF (0.5 mM), and $Na_3VO_4$ (0.5 mM) was added to the pellet and sonicated immediately. Protein amounts were adjusted using the Bradford assay (Bio-Rad). Proteasomal activity was measured in a reaction solution composing of 1/4 protein lysate, 1/4 reaction buffer (HEPES 100 mM, EGTA 10 mM, ATP 5 mM), and 2/4 fluorogenic substrate Suc-LLVY-AMC (Boston Biochem, 100 μM). Fluorescence was detected using an Infinite 200 PRO plate reader for 3 h at 5-min intervals at 37°C.

The effects of lactacystin on chymotrypsin-related proteasomal activity were also measured in living neurons by means of the fluorogenic proteasome substrate LLVY-Rhodamine-110 (LLVY-R110) as follows: At 16–19 days in culture, cortical neurons, growing within 6 mm glass cylinders, pairs of which were adhered to glass coverslips, were treated with lactacystin (10 μM) or carrier solution for 4 h, after which they were exposed to LLVY-R110 (Proteasome 20S Activity Assay Kit, Sigma; final dilution 1:800). Pairs of preparations were imaged side by side on an inverted confocal microscope for several hours at 5-min intervals as described below (Microscopy and image analysis).

## Measuring proteasomal inhibition using Ub-R-GFP

Cortical neurons growing within 8-mm glass cylinders on glass coverslips were transfected with destabilized GFP (Ub-R-GFP; Dantuma *et al*, 2000) in pEGFP-N1 (Clontech) with or without CFP (pECFP-C1, Clontech) by calcium phosphate transfection on days 9–10 *in vitro* as described previously (Bresler *et al*, 2004). Experiments were performed at 14–19 days in culture. After 1 h of baseline imaging (described below) at 10-min intervals, lactacystin (10 μM) or carrier solution was added to the preparation and imaging was continued for > 0.5 day at 10 min intervals.

## Pulsed SILAC

After cortical cells were grown for 14 days, media containing heavy isotope-labeled variants of lysine (Lys8-$^{13}C_6$, $^{15}N_2$) and arginine (Arg10-$^{13}C_6$, $^{15}N_4$) (Cambridge Isotope Laboratories) was added to the culture dishes, resulting in an excess of heavy lysine and arginine (5:1) as compared with the non-labeled variants of these amino acids. Neurons were harvested after 24 h by gently washing the cells three times in a physiological solution ("Tyrode's", 119 mM NaCl, 2.5 mM KCl, 2 mM $CaCl_2$, 2 mM $MgCl_2$, 25 mM HEPES, 30 mM glucose, buffered to pH 7.4), aspirating all the solution, and immediately adding 100 μl of lysis buffer composed of 10% SDS (Sigma), 30 mM Tris–HCl (Sigma), 3.4% glycerol, 25 mM DTT (Sigma), and 0.5% v/v protease inhibitor (Calbiochem). The cells were then scraped in the lysis buffer using a disposable cell scraper. The lysate was then collected, pipetted vigorously on ice, boiled for 5 min, and frozen at −80°C until used.

## Multiplexed dynamic SILAC

For multiplexed SILAC experiments, the cells were grown in lysine and arginine-free MEM (Biological Industries) to which "heavy" (H) variants (Lys8-$^{13}C_6$, $^{15}N_2$; Arg10-$^{13}C_6$, $^{15}N_4$) or "medium" (M) variants (Lys6-$^{13}C_6$; Arg6-$^{13}C_6$) were added such that final

concentrations (0.4 and 0.6 mM, respectively) were identical to the nominal lysine and arginine concentrations in our standard cell culture media. After 2 weeks, cells were gently washed by partial replacement of media with lysine and arginine-free media to which "light" lysine and arginine (Sigma) were added at 5× excess. At this time point, lactacystin (10 μM) or epoxomicin (5 μM) was added to one set of preparations. Cells were harvested after 4, 10, and 24 h, mixed together (as pairs of time-matched treated and control sets) and run on preparative gels. Each lane (including its stacking gel section) was then cut into five slices which were subjected separately to MS analysis. To exclude the possibility of a specific labeling type affecting the experimental outcome, one biological replicate was performed in which the labeling (heavy or medium type isotopic variants) was replaced between the treated and control samples.

### Fractional incorporation of H and M variants

To measure the degree to which proteins became labeled with H or M amino acids after 2 weeks of growth in the presence of these amino acids, cortical neurons were grown in culture for 2 weeks in either in H or M media (without proteasome inhibitors). Lysates of the preparations were then subjected to MS analysis; H/L or M/L ratios were measured separately for each identified peptide, and ratios were combined (averaged) for each protein for which ratios were obtained for at least 2 peptides. These ratios were then converted to fractional saturation values (i.e. values between 0 and 1.0) by calculating for each protein:

$$\text{Fractional saturation labeling} = \frac{H}{H+L} = \frac{R}{R+1} \qquad (3)$$

where $R$ is the H/L or M/L ratio calculated by MaxQuant (see below).

### Immunolabeling against synaptic proteins following proteasome inhibition

At 14 or 15 days in culture, primary hippocampal cultures prepared as described above were treated with 10 μM lactacystin (Santa Cruz Biotechnology) or vehicle. The cells were fixed after 10-h (Fig 2A and B), or at 2-h intervals over a time period of 24 h (Fig 2C and D), and stained using antibodies for synaptic proteins. Immunolabeling was performed by washing the cells in Tyrode's solution followed by fixation with 4% paraformaldehyde (PFA) and 120 mM sucrose in phosphate-buffered saline (PBS; fixative solution) for 20 min. For the 2-h interval experiments, 1% PFA in sodium acetate buffer (pH = 6) plus sucrose was used. PFA-fixed cells were permeabilized for 10 min in the appropriate fixative solution to which 0.25% Triton X-100 (Sigma) was added. The cells were washed three times in PBS, incubated in 10% bovine serum albumin (BSA, Sigma) for 1 h at 37°C or RT, and incubated overnight at 4°C with primary antibodies in PBS and 1% BSA. The cells were then washed three times for 5 min with PBS and incubated for 1 h at room temperature with secondary antibodies in PBS and 1% BSA. The cells were washed again with PBS three times and imaged immediately.

Primary antibodies included: mouse anti-synapsin I, 1:400 (TransLabs); mouse anti-PSD-95, clone 108E10, 1:200 (Synaptic Systems); rabbit anti-SV2A, 1:1,000 (Synaptic Systems); monoclonal

anti-Bassoon 1:400 (a generous gift of Craig Garner, DZNE, Berlin, Germany); guinea pig anti-ProSAP2, 1:800 (a generous gift of Tobias M. Boeckers, Ulm University, Germany) or rabbit anti-ProSAP2, 1:500 (Synaptic Systems; 2-h intervals experiments); mouse anti-RIM, 1:200 (BD Transduction Laboratories). Secondary antibodies included: Cy5 donkey anti-mouse (Jackson ImmunoResearch Laboratories), Cy5 donkey anti-guinea pig (Jackson ImmunoResearch Laboratories), Alexa Fluor 488 goat anti-rabbit (Molecular Probes). All secondary antibodies were used at a dilution of 1:200.

### In gel proteolysis and mass spectrometry analysis

Thirty micrograms of protein from each time point was sonicated, boiled, and separated on 4–15% SDS–PAGE (polyacrylamide gel electrophoresis). Each lane was sliced into five sections. The proteins in each gel slice were reduced with 3 mM DTT (60°C for 30 min), modified with 10 mM iodoacetamide in 100 mM ammonium bicarbonate (in the dark, room temperature for 30 min) and digested in 10% acetonitrile and 10 mM ammonium bicarbonate with modified trypsin (Promega) at a 1:10 enzyme-to-substrate ratio, overnight at 37°C. An additional second trypsinization was done for 4 h.

The resulting tryptic peptides were desalted using C18 tips (Harvard) dried and re-suspended in 0.1% formic acid. They were analyzed by LC-MS/MS using a Q Exactive Plus mass spectrometer (Thermo) fitted with a capillary HPLC (easy nLC 1000, Thermo). The peptides were loaded onto a homemade capillary column (25 cm, 75 μm ID) packed with Reprosil C18-Aqua (Dr Maisch GmbH, Germany) in solvent A (0.1% formic acid in water). The peptide mixture was resolved with a (5–28%) linear gradient of solvent B (95% acetonitrile with 0.1% formic acid) for 105 min followed by 15 min gradient of 28–95% and 15 min at 95% acetonitrile with 0.1% formic acid in water at flow rates of 0.15 μl/min. Mass spectrometry was performed in a positive mode (m/z 350–1,800, resolution 70,000) using repetitively full MS scan followed by collision-induced dissociation (HCD, at 35 normalized collision energy) of the 10 most dominant ions (> 1 charges) selected from the first MS scan. The AGC settings were $3 \times 10^6$ for the full MS and $1 \times 10^5$ for the MS/MS scans. The intensity threshold for triggering MS/MS analysis was $1 \times 10^4$. A dynamic exclusion list was enabled with exclusion duration of 20 s.

The mass spectrometry data were analyzed using MaxQuant software 1.5.1.2. (www.maxquant.org) for peak picking identification and quantitation using the Andromeda search engine, searching against the rat Uniprot database with mass tolerance of 20 ppm for the precursor masses and 20 ppm for the fragment ions. Oxidation on methionine, phosphorylation on STY, gly-gly on K, and protein N-terminus acetylation were accepted as variable modifications, and carbamidomethyl on cysteine was accepted as a static modification. Minimal peptide length was set to six amino acids, and a maximum of two miscleavages was allowed. Peptide- and protein-level false discovery rates (FDRs) were filtered to 1% using the target-decoy strategy. Protein tables were filtered to eliminate the identifications from the reverse database, and common contaminants and single peptide identifications. The data were quantified by SILAC analysis using the same software. H/L, M/L, and H/M ratios for all peptides belonging to a particular protein species were pooled, providing a ratio for each protein.

The mass spectrometry proteomics data have been deposited to the ProteomeXchange Consortium (http://proteomecentral.proteomexchange.org) via the PRIDE partner repository (Vizcaino *et al*, 2013) with data set identifiers PXD004711, PXD004726, PXD004813.

## Synaptic protein selection criteria

To examine the specific effects of proteasome inhibitors on a group of synaptic and synaptically related proteins, we collated a list of 314 proteins (Table EV4) that are either synapse-specific, highly enriched in synaptic compartments, or implicated in synaptic function (synaptic vesicle proteins, proteins involved in synaptic vesicle recycling, active zone proteins, neurotransmitter receptors, postsynaptic scaffolding molecules, adhesion molecules implicated in synaptic organization, and others). Of the proteins in this list, only those that were identified in all experiments and complied with the selection criteria for our data (e.g. minimum number of peptides, see details in "Data Analysis" below) were chosen for subsequent analysis.

## Normalization to long-lived proteins

As described above, multiplexed SILAC data samples were mixed together (as pairs of time-matched lactacystin-treated and control sets) and run together on preparative gels. To correct for potential remaining differences in the relative amounts of material collected from treated and control sets, we normalized all ratios to ratios measured for a group of seven abundant proteins previously shown to be particularly long-lived (Toyama *et al*, 2013; Table 1). Normalization was performed separately for each experiment and time point.

## Data analysis

For pulsed SILAC experiments, two separate experiments were performed, resulting in 1,409 proteins for which H/L ratios were obtained in both experiments and both conditions (H/L ratios based on at least two peptides in each experiment and condition). Data from both experiments were pooled, and the average H/L values were used thereafter.

Under certain assumptions, such H/L ratios can be used to estimate turnover rates of identified proteins (Cohen *et al*, 2013). To that end, average H/L ratios were first converted to fractional incorporation ratios ($F_t$) and corrected to the maximal expected ratio ($F_{max} \approx 5/(5 + 1)$) as follows:

$$F_t = \frac{1}{F_{max}} \cdot \frac{H_t}{H_t + L_t} = \frac{1}{F_{max}} \cdot \frac{R_t}{R_t + 1} \qquad (4)$$

where $R_t$ is the H/L ratio at time $t$.

Half-lifetime estimates, based on a single 24-h time point, were then calculated from $F_t$ according to:

$$F_t(t) = 1 - e^{-t/\tau} \qquad (5)$$

where $\tau$ is the turnover time constant. Solving for $\tau$ gives:

$$\tau = \frac{-t}{\ln(1 - F_t)} \qquad (6)$$

$\tau$ values were then converted to half-life ($t_{1/2}$) values according to:

$$t_{1/2} = \ln(2) \cdot \tau \qquad (7)$$

The apparent fold change in half-lifetime was then calculated by:

$$\text{fold change} = \frac{\tau_{\text{lacta,estimated}}}{\tau_{\text{control,estimated}}} \qquad (8)$$

For multiplexed SILAC, we performed three experiments using lactacystin and identified > 4,500 proteins. Data on 1,750 proteins were used in subsequent analyses, after selecting only proteins which met the following criteria: (i) H/M ratios were quantified for at least 2 peptides in each experiment, and (ii) such ratios were obtained in all three experiments at all four time points. We performed two additional experiments using epoxomicin. Here too, analysis was limited to proteins for which H/M ratios were based on at least 2 peptides in each experiment, and for which ratios were obtained in both experiments at all four time points. As described in the Results section, data obtained from experiments using both inhibitors were pooled, resulting in a stringent list of measurements for all proteins for all five experiments and all four time points (20 samples/protein). Proteins were included in the latter list if H/M ratio measurements were obtained for at least 2 peptides in each of at least 11 of the 12 lactacystin samples and at least 7 of the 8 epoxomicin samples (176 synaptic proteins, 1,595 proteins in total; Table EV3). Analyses and statistical tests were performed using Microsoft Excel and Perseus (http://www.coxdocs.org/doku.php?id = perseus:start).

## Microscopy and image analysis

Imaging of immunolabeled neurons in proteasome inhibition experiments was performed using a custom designed confocal laser scanning microscope (Tsuriel *et al*, 2006) using a 40×, 1.3 NA Fluar objective. Excitation was performed at 488 nm (Argon Laser) or 633 nm (Helium Neon Laser). Fluorescence emissions were read using a 500–550 nm band-pass filter (Chroma) or 650 nm long-pass filter (Semrock), respectively. Images were collected by averaging four or six frames at two to four focal planes spaced 0.8 μm apart. All data were collected at a resolution of 640 × 480 pixels, at 12 bits per pixel. Image analysis was performed using custom written software (OpenView) written by N.E.Z.. Analysis was performed on maximal intensity projections of 2 sections (or 4 for 2-h interval experiments), located 0.8 μm apart. Intensities of fluorescent puncta were measured by programmatically centering 9 × 9 pixel (~1.4 × 1.4 μm) regions of interest on individual puncta and obtaining the average fluorescence intensity in each region.

For live imaging in Ub-R-GFP experiments, automated, multisite time-lapse microscopy was performed using the inverted confocal microscope described above. Excitation of CFP and Ub-R-GFP was performed at 457 nm and 488 nm (Argon Laser lines), respectively. CFP fluorescence emission was read using a 467–493 nm band-pass filter (Semrock). GFP emissions were split between two photomultipliers using a 550 dichroic mirror and filtered at 500–550 nm (GFP; Chroma) and 570–610 nm (nonspecific fluorescence; Chroma). Spectral unmixing was then used to remove the contributions of nonspecific fluorescence sources to Ub-R-GFP data. Images were

collected by averaging six frames at three focal planes spaced 0.8 μm apart. All data were collected at a resolution of 640 × 480, at 12 bits per pixel. Preparations were imaged within their cloning cylinders in the presence of their growth media (no perfusion). The objective was heated to 36°C, and a filtered mixture of 95% air and 5% CO$_2$ was continuously streamed into a custom enclosure placed over the cells throughout the duration of the experiment.

Live imaging of LLVY-R110 accumulation in living neurons was done in a similar manner. R110 fluorescence was recorded by exciting the preparations at 488 nm and collecting the emissions at 500–550 nm.

Long-term effects on neuronal morphology were measured as follows: Cortical neurons, growing on glass-bottomed plates, were infected with lentiviral vectors driving the expression of the postsynaptic density protein PSD-95 fused to mTurquoise2 and synapsin I fused to EGFP (on consecutive days, at 2–3 days *in vitro*, respectively). At 16 days in culture, cells were exposed to lactacystin (10 μM) and imaged immediately, after 24 h and after 48 h. In between imaging sessions, the cells were kept in a 37°C, 5% CO$_2$ humidified incubator. Excitation of mTurquoise2 and GFP was performed at 457 and 488 nm, respectively. mTurquoise2 fluorescence emission was read using a 467–493 nm band-pass filter. GFP emission was read using a 500–550 nm band-pass filter (Chroma). Differential interference contrast (DIC) images were collected and used to assess overall neuronal integrity and morphology. Images were collected by averaging six frames at eight focal planes spaced 0.8 μm apart. Imaging resolution and environmental conditions were as described above.

### Recordings of network activity

Cortical neurons were plated on thin glass multielectrode array (MEA) dishes containing 59, 30 μm diameter, electrodes arranged in an 8 × 8 array, spaced 200 μm apart. The dishes were covered by a custom designed cap containing a submerged platinum wire loop serving as a ground electrode, heated to 37°C, and provided with a filtered stream of 5% CO$_2$ and 95% air through an inlet in the cap. Network activity was recorded through a commercial 60-channel headstage/amplifier (Inverted MEA1060, MCS) with a gain of 1,024× and frequency limits of 1–5,000 Hz. The amplified signal was further amplified and filtered using a bank of programmable filter/amplifiers (Alpha-Omega), multiplexed into 16 channels, and then digitized by two A/D boards (Microstar Laboratories) at 12 KSamples/sec per channel. Data acquisition was performed using AlphaMap (Alpha-Omega). All data were stored as threshold crossing events with the threshold set to −20 μV. Electrophysiological data were imported to Matlab (MathWorks) and analyzed using custom written scripts. To determine the effects of proteasome inhibitors on neuronal network activity, cortical neurons were grown for 17 days on thin glass MEAs. After 5 h of baseline recording, lactacystin was applied at a concentration of 10 μM. Recordings were then continued for another 20 h.

### Western blots

Cortical cells were washed using Tyrode's solution and lysed in RIPA buffer or 8M urea, 100 mM Tris–HCl. Protein concentrations were measured by the Bradford assay (Hershko *et al*, 1982) using BSA as the standard. Equal protein amounts (30 or 40 μg) were separated by SDS gel electrophoresis and transferred to PVDF membrane. Staining was performed using anti-P21 antibody (BD Pharmingen, 1:500), anti-α tubulin (Millipore, 1:10,000), or anti-Ub conjugates (an in-house developed antibody; Gonen *et al*, 1999, 1:1,000) as primary antibodies and peroxidase-conjugated (ImmunoResearch Laboratories) as secondary antibody. Enhanced chemiluminescence (ECL) (Pierce) was used for immunodetection.

**Expanded View** for this article is available online.

## Acknowledgements

We are grateful to Nico Dantuma for the provision of the Ub-X-GFP constructs, to Aaron Ciechanover and Avram Hershko for many helpful suggestions and discussions and to Ido Livne, Larisa Goldfeld, Leonid Odesski, and Arava Fisher-Lavie for their invaluable assistance. This work was supported by funding from the DFG German-Israeli Foundation DIP (RO3971/1-1), the Deutsche Forschungsgemeinschaft (SFB 1089 - Synaptic Micronetworks in Health and Disease), the Israel Science Foundation (ISF) (1175/14), ISF European Community's Seventh Framework Programme FP7/2012 (TreatPolyQ, Grant Agreement No. 264508), and the Allen and Jewel Prince Center for Neurodegenerative Processes of the Brain.

## Author contributions

VH, LDC, and RZ carried out the experiments. LDC, VH, TZ, and NEZ performed the analysis. LDC, VH, TZ, and NEZ wrote the manuscript.

## Conflict of interest

The authors declare that they have no conflict of interest.

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
