## [Review Process File · The EMBO Journal]

Manuscript EMBO-2015-93594

The Effects of Proteasomal Inhibition on Synaptic Proteostasis

Vicki Hakim, Laurie Cohen, Rina Zuchman, Tamar Ziv and Noam Ziv

Corresponding author: Noam Ziv, Technion

Review timeline:	Submission date:	08 December 2015
	Editorial Decision:	08 January 2016
	Resubmission:	01 June 2016
	Editorial Decision:	18 July 2016
	Revision received:	02 August 2016
	Accepted:	08 August 2016

Editor: Karin Dumstrei

Transaction Report:

1st Editorial Decision

08 January 2016

Thank you for submitting your manuscript to The EMBO Journal. Your study has now been seen by three referees and I am afraid that the overall conclusion is not a positive one.

While the referees appreciate the interest of the topic, they also raise concerns with the analysis that preclude publication here. The referees raise concerns primarily regarding the conclusiveness of the analysis. They find that the dataset is essentially negative data, that there are possible confounding issues (like lactacystine toxicity) and that there are no positive controls. Referee #2 also finds that the extent of the analysis is too narrow and that one should have looked at the more proteins.

Given these comments from good experts in the field, I am afraid that I can't offer to consider publication here. I am very sorry that I can't be more helpful on this occasion, but I hope that you find the referees' comments helpful.

REFEREE COMMENTS

Referee #1:

The authors analyze here the rates of synaptic protein degradation in cultured neurons, under proteasome inhibition. They use rely on mass spectrometry and SILAC approaches, and study the effects of two proteasome inhibitors, lactacystin and epoxomicin. They find that the degradation rates of most synaptic proteins are not affected by the two drugs, albeit the degradation of a handful of proteins was significantly slowed by proteasome inhibition. The authors conclude that most synaptic proteins are not degraded by the ubiquitin-proteasomal system.

In the current form, it is difficult to accept the conclusions of the manuscript. They are essentially negative, and therefore difficult to interpret. A number of controls are required, to raise the level of confidence of the conclusions:

The authors require a positive control. They need to express a protein tagged with a specific degron sequence, which directs it to the proteasome, and they need to show that the degradation of such a protein stops upon lactacystin or epoxomicin treatment. First, this type of experiment needs to be performed by quantitative immunoblotting, to determine the amounts of the protein of interest with higher precision than it is possible for mass spectrometry. Second, the authors should also perform the analysis by mass spectrometry, using both approaches presented in their manuscript, in order to calibrate their measurements to a known and controlled protein (the degron-tagged protein). The blot in Supplementary Figure 1 is not sufficient to show that lactacystin blocks the proteasome completely. It only shows a certain increase in the amount of ubiquitin conjugates. But is the inhibition complete? The fact that no effect is seen on neuronal spiking (Supplementary Figure 5) actually argues against this hypothesis. Other groups have observed severe lactacystin-induced cytotoxicity, even at lower concentrations than those used by the authors here (for example, Perez-Alvarez et al., 2009, British Journal of Pharmacology). The lack of any measurable effects in Supplementary Figure 5 may be attributed to an insufficient penetration of lactacystin in the cells. This issue could be resolved by a proper positive control, as suggested in the previous paragraphs. If the results from Supplementary Figure 5 are indeed true, and the neurons are not negatively affected by proteasome inhibition, the authors should probably prolong the exposure to lactacystin, in order to obtain a clearer image of the protein degradation. However, if a positive control (see above) looks convincing even for the short time periods used in the manuscript, this issue may be irrelevant.

Do the authors note any changes in other degradation machineries? A few immunostaining experiments (for, among others, lysosomal markers) would be important to add.

Referee #2:

In this manuscript, the authors attempted to globally explore synaptic protein turnover by the latest mass spec approach. Recently, metabolic labeling combined with high throughput mass spec has emerging as a popular method to analyze protein turnover (protein synthesis and degradation). The authors adopted the strategy for neuronal culture and carefully characterized the turnover of less than 2000 proteins upon the inhibition of proteasomal degradation. The experiments were well planned (e.g. multiplex SILAC with 3 replicates) and the MS data analysis was solid (e.g. the use of at least 2 peptides for ID). Unfortunately, the authors found that only a small percentage of proteins were affected by the treatment and suggested that "most synaptic proteins are degraded through alternative pathways". To this reviewer, the conclusion is overstated. Main points are as follows.

1. Upon proteasomal inhibition, it is assumed that the fast turnover proteins would be accumulated rapidly, whereas the slow turnover proteins may appear to be not influenced, even if these are still degraded by the proteasomal pathway. Moreover, the long-term proteasomal inhibition will result in the blockage of protein synthesis, which will further reduce the effect on protein abundance. This scenario also occurs under the inhibition of protein synthesis. Therefore, it is very reasonable to see only a small portion of proteins are affected. If the author claimed that alternative pathways are

responsible. Then the major alternative pathway would be autophagy-lysosomal pathways, which can be interfered with specific chemicals. The authors should validate this possibility by other inhibitors or blocking all known degradation pathways simultaneously.

2. The scope of the analysis is relatively shallow. Considering that 12-14K proteins are expressed in neurons, this analysis only measured the most abundant ~2000 proteins. As the abundant proteins are often degraded at low rate, this could also explain why so few proteins are shown to be affected by proteasomal inhibition, as many fast turnover proteins are not identified in the analysis.

3. Based on other MS-base turnover studies, protein localization may also affect protein degradation. The authors used the whole cell lysate during the analysis. It is possible that the authors observed the sum effect of degradation in synaptic regions, neurites and soma.

4. The main methodology is largely published by other papers and thus is not novel.

Overall, it is an interesting study but with overstated interpretation. The limitation of the scope of MS analysis and the use of only proteasomal inhibition conditions prevent the full, unbiased understanding of protein degradation events in synapse.

Referee #3:

Hakim et al. examine the rates of synthesis and degradation of proteins in primary neurons in extended culture that were treated with proteasome active-site inhibitors (usually lactacystin). Most of the analysis uses quantitative SILAC-based mass spectrometry. Surprisingly, relatively few short-lived synaptic proteins become more long-lived in the presence of lactacystin, but there are substantial effects on protein synthesis rates. This latter effect is suggested to result from a proteotoxic stress response caused by proteasome inhibition, which is likely related to the unfolded protein response in the ER. This would be consistent with previous work on proteasome inhibition and the UPR.

Besides the unexpectedly weak degradation effects observed, this manuscript is noteworthy for its careful consideration of experimental assumptions and variables and its quantitative treatment of the data. This has the potential for clarifying conflicting observations in the literature on synaptic protein turnover. Many of the inferences are reasonable, albeit indirect.

On the negative side, the analysis was not done in any condition that directly alters, or monitors, synaptic function, so it is not yet clear to what extent the meager degradative changes observed from pharmacological inhibition of the proteasome in cultured rat cortical neurons extend to neurons undergoing excitation or inhibition (or to the intact brain). Importantly, left untested was how severely the proteasome was actually inhibited under the conditions used. I also felt that the paper was longer than it needed to be, and a number of the figures could be put in the Supplementary Materials.

Overall, I would like to see additional controls that support the central, essentially negative finding of minimal proteolytic perturbation by proteasome inhibition of these neurons.

Specific additional comments:

1. Proteasomes might only be inhibited weakly by the inhibitor treatments used, and if strong inhibition is not occurring, the major (negative) findings go out the window. Evidence for some inhibition is a) an anti-ubiquitin immunoblot (poorly documented; was also done with cortical neurons, not the hippocampal neurons used for microscopy), which shows more HMW ubiquitin with time of treatment (Fig. S1), b) the proteotoxic stress response, and c) the observed stabilization of certain proteins, such as NEDD4. None of these data, however, indicates whether the proteasomal protease activity is blocked by 10% or 100% or whether only a subset of the three different proteasome active sites (possibly mostly just the beta5 subunit sites) is blocked. Specific proteasomal peptidase activities can be monitored pretty sensitively in cell extracts, or even better, extracts from synaptosome preparations. I am not familiar with these cells/tissues, but it may be necessary to fractionate proteasomes on gradients if other peptidases are interfering with the assay.

This control is essential.

2. Another issue, which the authors do discuss in detail, is that inhibitor treatment was maximally 24 h long, and it might be hard to detect small changes in levels of proteins with longer half-lives. I wonder if it would be possible to monitor at longer times? Eventually, the cells would be expected to apoptose with inhibitor present; when does this start to happen with 10 μ M lactacystin treatment of these cells?

3. p. 4, bottom. Lactacystin is known to inhibit at least one additional cellular protease at micromolar concentrations - Cathepsin A.

4. Figure 4 at least should go to Supplemental. Figures 2 and 7 would also be candidates for moving.

5. p. 15. The UPS inhibition causes more than a response just in the ER (UPR). I don't believe proteasome upregulation is observed, for example, when the UPR is specifically induced by an ER stress.

6. The paper is very clearly written, but the writing could be made much more succinct.

7. What is meant by the word "Intact" in the title? Was proteasome "intactness" ever assessed?

8. Fig. S1. The blot needs to be labeled better (looks like it came from someone's group meeting) and should include a parallel vehicle control (DMSO?).

9. The correlation between the two inhibitors ($r=0.5$) is actually not that high. The authors should comment on this.

We truly appreciate the constructive comments made by all three reviewers, and did our best to address these.

Referee #1:

In the current form, it is difficult to accept the conclusions of the manuscript. They are essentially negative, and therefore difficult to interpret. A number of controls are required, to raise the level of confidence of the conclusions:

The authors require a positive control. They need to express a protein tagged with a specific degron sequence, which directs it to the proteasome, and they need to show that the degradation of such a protein stops upon lactacystin or epoxomicin treatment. First, this type of experiment needs to be performed by quantitative immunoblotting, to determine the amounts of the protein of interest with higher precision than it is possible for mass spectrometry. Second, the authors should also perform the analysis by mass spectrometry, using both approaches presented in their manuscript, in order to calibrate their measurements to a known and controlled protein (the degron-tagged protein).

The blot in Supplementary Figure 1 is not sufficient to show that lactacystin blocks the proteasome completely. It only shows a certain increase in the amount of ubiquitin conjugates. But is the inhibition complete? The fact that no effect is seen on neuronal spiking (Supplementary Figure 5) actually argues against this hypothesis. Other groups have observed severe lactacystin-induced cytotoxicity, even at lower concentrations than those used by the authors here (for example, Perez-Alvarez et al., 2009, British Journal of Pharmacology). The lack of any measurable effects in Supplementary Figure 5 may be attributed to an insufficient penetration of lactacystin in the cells. This issue could be resolved by a proper positive control, as suggested in the previous paragraphs.

If the results from Supplementary Figure 5 are indeed true, and the neurons are not negatively affected by proteasome inhibition, the authors should probably prolong the exposure to lactacystin, in order to obtain a clearer image of the protein degradation. However, if a positive control (see above) looks convincing even for the short time periods used in the manuscript, this issue may be irrelevant.

These are important points, which were raised by other reviewers as well.

Proteasomal inhibition:

We completely acknowledge that a clear demonstration and quantification of the extent to which lactacystin and epoxomicin inhibit proteasomal activity in these preparations should have been included. We originally assumed, naively perhaps, that the extensive literature on these inhibitors, the fact that their activities are rarely verified in most papers in which they are used and the fact that they are considered to be “gold standard” proteasome inhibitors were sufficient reasons to assume that proteasome function was strongly inhibited in our experiments. In retrospect this was a glaring omission, one pointed to by all three reviewers. We thus addressed this matter in four ways. We first used an *in-vitro* assay and a fluorogenic substrate to measure proteasomal activity in extracts from neurons treated with either

lactacystin or epoxomicin for 4 hours before they were washed and extracted. We find that these treatments resulted in ~20 fold reductions in fluorescence accumulation rates as compared to extracts of untreated neurons (Fig. S1). In the second, we repeated these experiments in living cells, using a similar fluorogenic substrate. Here we observed a ~3 fold reduction (Fig. S1), but this is almost certainly an underestimate, as we noted that substrate penetration into neurons was slow, and that fluorescence increases were observed only within intracellular organelles, indicating that the substrate, once cleaved, was not retained in the cytosol. Third, we measured by Western blots how lactacystin or epoxomicin affect cellular levels of p21^{waf1/cip1} a short-lived protein whose proteasomal degradation has been well characterized, finding that in the presence of these inhibitors, p21^{waf1/cip1} levels increased from practically undetectable to very substantial over 4-8 hours (Fig. S2). Finally, we performed live imaging using destabilized GFP (Ub-R-GFP) and found that lactacystin increases GFP accumulation rates ~20 fold while having no effects on a co-expressed fluorescent protein (CFP), both of which were expressed from identical expression vectors (Fig. 1). In sum, these data show that proteasomal inhibition in our experiments was very effective. These experiments are now described in the first section of the Results.

Several further comments in this regard –

- The blot in Supplementary Figure 1 (now Supplementary Figure 10), was never meant to calibrate inhibitor efficacy, only to verify that inhibitor stocks we purchased were pharmacologically active (as we originally wrote). This was done as a precaution for every stock of lactacystin and epoxomicin we purchased. This matter is now clarified in the text which was moved to Materials and Methods.
- The basis for the statement that quantitative immunoblotting is more precise than mass spectrometry is not clear to us. We are not aware of an established comparison which points to this conclusion. We note that our data (Fig. 6F) suggests that in our assays, quantification errors greater than 9% (2 standard deviations) will happen by chance for <5% of such measurements, and quantification errors greater than 14% (3 standard deviations) will happen by chance for <1% of measurements (a conclusion based on measurements made for 1,469 proteins). For biochemical assays aimed at comparing protein abundance in two samples, this is quite remarkable precision.

It is also important to emphasize that western blots provide information on total protein abundance; thus, changes in staining intensity, unlike the proteomic approach described in Fig. 5, do not reveal if such changes stem from altered synthesis or altered degradation, complicating interpretation of changes in protein abundance observed in western blots.

Inhibitor toxicity

Obviously, blocking the proteasome for prolonged periods is ultimately toxic. We did not mean to imply otherwise. We now explain this in the Discussion and add data suggesting that while detrimental effects are not readily observed during the first 24 hours they become quite obvious after 48 hours (Fig. S9). This is in line with the lack of obvious effects on network activity (Fig. S8), and the normal morphology of neurons which have been exposed to lactacystin for 10-24 hours (Figs. 1A,C; Fig.2) even when proteasomal activity in the same cells is effectively

suppressed (Figs. 1A,C). This is in perfect agreement with the study of Dantuma et al., 2000 (*Nat. Biotechnol.* **18**:538–43,) which reported that “*It is noteworthy that, whereas the cytotoxic effect of proteasome inhibitors requires between 24 and 48 h, a significant accumulation of the GFP reporter was already evident after 2h*”. This being so, extending the analysis beyond 24 hours did not seem prudent.

Interestingly, the paper mentioned above by the reviewer implicates mitochondria in lactacystin-induced cell death whereas our own data suggests that proteins most strongly affected by proteasomal inhibition were related to mitochondria (Results).

Do the authors note any changes in other degradation machineries? A few immunostaining experiments (for, among others, lysosomal markers) would be important to add.

Our proteomic analysis did not point to obvious changes in alternative catabolic pathways. However, we are now fiercely examining alternatives using both proteomic and cell-biological approaches. We feel that a serious consideration of other catabolic pathways is beyond the scope of this manuscript (which is quite long already – 10 primary and 10 supplementary figures).

Referee #2:

1. Upon proteasomal inhibition, it is assumed that the fast turnover proteins would be accumulated rapidly, whereas the slow turnover proteins may appear to be not influenced, even if these are still degraded by the proteasomal pathway. Moreover, the long-term proteasomal inhibition will result in the blockage of protein synthesis, which will further reduce the effect on protein abundance. This scenario also occurs under the inhibition of protein synthesis. Therefore, it is very reasonable to see only a small portion of proteins are affected. If the author claimed that alternative pathways are responsible. Then the major alternative pathway would be autophagy-lysosomal pathways, which can be interfered with specific chemicals. The authors should validate this possibility by other inhibitors or blocking all known degradation pathways simultaneously.

We completely agree that proteins with faster turnover rates will be the proteins most strongly impacted, and that proteins with particularly slow turnover rates might be missed. In fact, this is one of the major points our manuscript makes. We therefore introduced known protein turnover rates to examine this matter in a quantitative manner (Figs. 9,10) within an analytical framework we developed for this purpose (Fig. 8), and went to great lengths to characterize our sensitivity and the range of turnover rates within which our data provides meaningful information (Fig. 6). We find that for proteins with half-lives < 5 days our data are informative. We think that this matter was therefore fully addressed.

As to the effects of blockage of protein synthesis and their effects on protein abundance, the experiments of Figs. 5-10 – the major part of the paper – were designed in such a manner that

changes in protein synthesis would have no effect on our readouts. (Fig. 5). To further clarify this crucial point we added a new illustration (Fig. 5B).

2. The scope of the analysis is relatively shallow. Considering that 12-14K proteins are expressed in neurons, this analysis only measured the most abundant ~2000 proteins. As the abundant proteins are often degraded at low rate, this could also explain why so few proteins are shown to be affected by proteasomal inhibition, as many fast turnover proteins are not identified in the analysis.

We obtained information on the effects on 1,416 proteins and 174 synaptic proteins, including many well-studied ones. We are not aware of any comparable large scale survey of synaptic protein metabolism sensitivity to proteasomal inhibition that is even remotely close in scope, not to mention the fact that in prior papers concerning synaptic proteins, no separation was made between effects on protein degradation and synthesis. We thus feel that the claim that our analysis is shallow is debatable.

In fairness, however, we acknowledge that MS based analyses are inherently biased toward the more abundant proteins. On the other hand, we note that the proteomic findings are generally in agreement with the limited survey we did using immunocytochemistry (Fig. 2) where only one of six proteins showed evidence for UPS-mediated degradation under basal conditions.

In this regard it is also worth considering the results of two other large scale proteomic studies carried out in human Jurkat cells (Udeshi et al., 2012; *Mol Cell Proteomics* **11**:148-59) and U2OS osteosarcoma cells (Larance et al., 2013; *Mol Cell Proteomics* **12**:638-50). These noted that

- *“Steady-State Protein Levels are Largely Unaffected Following Proteasome and DUB Inhibition ... We were initially surprised to find that nearly all protein levels were unaffected following MG-132 or PR-619 treatment”* (Udeshi et al., 2012, ~2,500 proteins, MG132 or PR-619 (a DUB inhibitor), 5 hour incubations)
- *“A particularly striking finding of this study on U2OS cells was that inhibition of the proteasome for 6 h did not cause an increase in the abundance of almost any of the 5000 proteins identified”* (Larance et al., 2013; MG132, 6 hour incubations)

Explanations were provided that could partially account for these findings (low stoichiometry of ubiquitinated protein forms, protein synthesis shutdown due to an unfolded protein response); Nevertheless, these extensive surveys do indicate, that UPS-mediated degradation under basal conditions might not be as widespread as commonly believed, which is one of the tentative conclusions we reach on the matter of synaptic protein degradation. Moreover, in our experiments 1) potential effects on protein synthesis were largely eliminated by protocol design, and 2) we did identify proteins for which degradation was slowed down in the presence of proteasomal inhibitors in a manner which agreed quite well with their known half-lives.

3. Based on other MS-base turnover studies, protein localization may also affect protein degradation. The authors used the whole cell lysate during the analysis. It is possible that the authors observed the sum effect of degradation in synaptic regions, neurites and soma.

We completely agree; here too, we note, however, that our immunocytochemical analysis (Fig. 2) did not provide any indication for more pronounced UPS-mediated degradation at synapses. On the other hand, our evidence points to the importance of proteasomal degradation outside of synapses, possibly as part of quality control processes, or perhaps as regulatory steps *en route* to synapses (see also Saliba et al., 2007 *J. Neurosci.* **27**: 13341–51). An interesting study in this regard (Shin et al., 2012; *Nat. Neurosci.* 15: 1655–66) concerns GKAP, a postsynaptic scaffolding molecule previously shown to be degraded via the UPS. Here it was found that over-excitation decreases GKAP levels at synapses; yet this decrease was not prevented by proteasomal inhibitors – instead, these induced large GKAP aggregates in the soma and proximal dendrites, suggesting centralized degradation sites. This and similar findings show that it is imperative to examine the entire cellular contents even where synaptic proteins are concerned, in order to end up with correct interpretations of proteasomal function in synaptic protein metabolism.

4. The main methodology is largely published by other papers and thus is not novel.

We have not claimed that the main methodologies used here are novel. Our major goal was to examine how synaptic protein metabolism is affected by proteasomal inhibition, and the proteomic approaches described here were means to this end. Having said this, we are not aware of any study to date that has used such methods in conjunction with information on protein half-lives (Figs. 9-10) within the quantitative framework we devised (Fig. 8) to examine the findings validity and place boundaries on their interpretation.

Referee #3:

Besides the unexpectedly weak degradation effects observed, this manuscript is noteworthy for its careful consideration of experimental assumptions and variables and its quantitative treatment of the data. This has the potential for clarifying conflicting observations in the literature on synaptic protein turnover. Many of the inferences are reasonable, albeit indirect. On the negative side, the analysis was not done in any condition that directly alters, or monitors, synaptic function, so it is not yet clear to what extent the meager degradative changes observed from pharmacological inhibition of the proteasome in cultured rat cortical neurons extend to neurons undergoing excitation or inhibition (or to the intact brain). Importantly, left untested was how severely the proteasome was actually inhibited under the conditions used. I also felt that the paper was longer than it needed to be, and a number of the figures could be put in the Supplementary Materials.

Overall, I would like to see additional controls that support the central, essentially negative finding of minimal proteolytic perturbation by proteasome inhibition of these neurons.

1. Proteasomes might only be inhibited weakly by the inhibitor treatments used, and if strong inhibition is not occurring, the major (negative) findings go out the window. Evidence for some

inhibition is a) an anti-ubiquitin immunoblot (poorly documented; was also done with cortical neurons, not the hippocampal neurons used for microscopy), which shows more HMW ubiquitin with time of treatment (Fig. S1), b) the proteotoxic stress response, and c) the observed stabilization of certain proteins, such as NEDD4. None of these data, however, indicates whether the proteasomal protease activity is blocked by 10% or 100% or whether only a subset of the three different proteasome active sites (possibly mostly just the beta5 subunit sites) is blocked. Specific proteasomal peptidase activities can be monitored pretty sensitively in cell extracts, or even better, extracts from synaptosome preparations. I am not familiar with these cells/tissues, but it may be necessary to fractionate proteasomes on gradients if other peptidases are interfering with the assay. This control is essential.

These control experiments were now performed and provided as Fig. 1, S1 and S2. Please see our response to reviewer #1 (“Proteasomal inhibition”) for a full explanation.

2. Another issue, which the authors do discuss in detail, is that inhibitor treatment was maximally 24 h long, and it might be hard to detect small changes in levels of proteins with longer half-lives. I wonder if it would be possible to monitor at longer times? Eventually, the cells would be expected to apoptose with inhibitor present; when does this start to happen with 10 uM lactacystin treatment of these cells?

Please see our response to reviewer #1 (“Inhibitor toxicity”).

3. p. 4, bottom. Lactacystin is known to inhibit at least one additional cellular protease at micromolar concentrations - Cathepsin A.

We thank the reviewer for this comment. We now mention this point when introducing epoxomicin.

4. Figure 4 at least should go to Supplemental. Figures 2 and 7 would also be candidates for moving.

Fig. 4B was moved to a supplementary figure (S6); Figs. 2B,C were moved to a supplementary figure (S3). We added, however, a new illustration to figure 4 (now Fig. 5B) as it seemed that the underlying logic of the experiments of Figs. 5 – 10 was not sufficiently clear.

5. p. 15. The UPS inhibition causes more than a response just in the ER (UPR). I don't believe proteasome upregulation is observed, for example, when the UPR is specifically induced by an ER stress.

The reviewer is probably correct, and it is likely that an UPR is only one of the cellular responses evoked by proteasome inhibition. We therefore rephrased the text concerning this matter to be more inclusive and removed the mention of an UPR from the abstract.

We note that the conclusion that proteasomal inhibition invokes an UPR (among others) was also reached in the proteomic study of Larance et al. (2013), a conclusion they further substantiated by measuring eIF2alpha phosphorylation levels and establishing that these were increased as expected.

6. The paper is very clearly written, but the writing could be made much more succinct.

We tried to shorten it where possible, although perhaps it could be shortened further. Please note, however, that it was originally pointed out by the editor that the terminology (H/M, H/L, etc.) can be confusing and that it is easy to get lost in technicalities. We therefore tried to be as clear as possible, at the expense of brevity.

7. What is meant by the word "Intact" in the title? Was proteasome "intactness" ever assessed?

What we were referring to is intact function, that is, intact in the sense that proteasome function is unperturbed / undisturbed

As this term seems to be confusing, we shortened the title to "Dependence of Synaptic Protein Degradation on Proteasomal Function". We also rephrased the text in all other instances in which the term 'intact' was used.

8. Fig. S1. The blot needs to be labeled better (looks like it came from someone's group meeting) and should include a parallel vehicle control (DMSO?).

We apologize for this. As we mentioned in our response to reviewer #1, this blot was merely an example of an assay we did as a precaution whenever we purchased a new batch of lactacystin or epoxomicin, just to verify that the material was pharmacologically active. This matter is now clarified in the text, which was moved to Materials and Methods.

9. The correlation between the two inhibitors ($r=0.5$) is actually not that high. The authors should comment on this.

More information on this matter as well as several additional details are now provided in the legend of this figure (S7).

Thanks for submitting your revised manuscript to The EMBO Journal. Your study has now been seen by the original referees #1 and 3.

As you can see below, the referees appreciate the introduced changes and support publication here. There are a few minor comments that I would appreciate your response to in the point-by-point response and perhaps in manuscript text if helpful.

Referee #2 also finds the text a bit long. Have a careful look at the manuscript and see if it makes sense to move some parts into the appendix.

 REFEREE REPORTS

Referee #1:

The authors replied to my comments by performing several experiments. My most important comment was that the authors need to show that they inhibit the proteasome completely with lactacystin and epoxomicin.

I think the authors have done what could be done in this direction. I am still not fully convinced that complete proteasomal inhibition has been obtained. For example, in Figure 1 the authors show that the amounts of a destabilized GFP increase after treatment with lactacystin treatment, indicating that at least some proteasome inhibition takes place. But why is there no change in the concentration of CFP, which is co-expressed in these cultures? Is CFP not degraded by the proteasome? But, since we do not know what happens with CFP in the neurons, this type of argument is not a solid one. And the authors' experiments have show that there is at least a substantial degree of proteasome inhibition. Therefore, I now support the publication of the manuscript.

Referee #3:

The revision from Hakim et al. addresses many of the concerns I had with the original submission, which generally speaking I had liked already. The results, again, have two major take-home messages. One highlights the fact that synaptic proteins are mostly fairly stable and not strongly affected by proteasome inhibitors and even for short-lived proteins, relatively few seem to be strongly affected by these inhibitors. The second is that protein synthesis of synaptic proteins is strongly affected, probably through the UPR (or more generally, the Integrated Stress Response, which is worth stating). The relatively small number of proteins that are turning over rapidly due to proteasomal degradation may surprise some, but the analysis is careful and I agree with the authors.

I still worry that the lack of strong effects of lactacystin and epoxomicin is due to inhibition only of the chymotrypsin-like active sites (such findings have been made in yeast, for example). However, the authors do now check in vivo R-GFP, an N-end rule substrate, which was strongly impaired by these drugs in neurons (they also mentioned UFD substrates, but I don't think they actually checked any of these). Overall, I think they do a good job of considering various variables, and I think this is worth publishing in the EMBO Journal. It still feels incredibly long, and some thought should still be given on parts to put in the Supplementary Data.

Referee #1:

The authors replied to my comments by performing several experiments. My most important comment was that the authors need to show that they inhibit the proteasome completely with lactacystin and epoxomicin.

I think the authors have done what could be done in this direction. I am still not fully convinced that complete proteasomal inhibition has been obtained. For example, in Figure 1 the authors show that the amounts of a destabilized GFP increase after treatment with lactacystin treatment, indicating that at least some proteasome inhibition takes place. But why is there no change in the concentration of CFP, which is co-expressed in these cultures? Is CFP not degraded by the proteasome? But, since we do not know what happens with CFP in the neurons, this type of argument is not a solid one. And the authors' experiments have show that there is at least a substantial degree of proteasome inhibition. Therefore, I now support the publication of the manuscript.

We thank the reviewer for his support. Indeed, the degree of inhibition was quantified by comparing the rate of destabilized GFP or fluorogenic substrate accumulation in the presence and absence of inhibitors (Figs. 1, EV1). These comparisons led us to conclude that the inhibitors reduced degradation rates ~20 fold, although probably not completely. Note, however, and as we show in Fig. 8B, that any suppression beyond a 10x suppression would result in essentially the same $\log_2(H/M)$ ratio readouts; thus differences between 20 fold and complete suppressions of proteasomal activity would be nearly negligible in the assays used here.

As to CFP, EGFP and its spectral variants are very stable proteins, with half-lives >24 hours when expressed in mammalian cell lines¹ (such as CHO cells), and probably longer in neurons, in which turnover rates are generally slower, even for the same proteins². It is therefore not very surprising that ECFP concentrations were quite stable over the time course of these experiments.

Referee #3:

The revision from Hakim et al. addresses many of the concerns I had with the original submission, which generally speaking I had liked already. The results, again, have two major take-home messages. One highlights the fact that synaptic proteins are mostly fairly stable and not strongly affected by proteasome inhibitors and even for short-lived proteins, relatively few seem to be strongly affected by these inhibitors. The second is that protein synthesis of synaptic proteins is strongly affected, probably through the UPR (or more generally, the Integrated Stress

¹ Li et al., (1998) Generation of destabilized green fluorescent protein as a transcription reporter. J Biol Chem. 1998 Dec 25;273(52):34970-5.

² Price et al., (2010) Analysis of proteome dynamics in the mouse brain. Proc Natl Acad Sci U S A. 107:14508-13.

Response, which is worth stating). The relatively small number of proteins that are turning over rapidly due to proteasomal degradation may surprise some, but the analysis is careful and I agree with the authors.

I still worry that the lack of strong effects of lactacystin and epoxomicin is due to inhibition only of the chymotrypsin-like active sites (such findings have been made in yeast, for example). However, the authors do now check in vivo R-GFP, an N-end rule substrate, which was strongly impaired by these drugs in neurons (they also mentioned UFD substrates, but I don't think they actually checked any of these). Overall, I think they do a good job of considering various variables, and I think this is worth publishing in the EMBO Journal. It still feels incredibly long, and some thought should still be given on parts to put in the Supplementary Data.

We thank the reviewer for his positive assessment.

- Integrated Stress Response (ISR): Looking into our data, we found evidence that expression levels of four proteins, whose expression was reported to be elevated as part of an ISR in an ATF4-dependent manner³, were elevated following exposure to lactacystin. The expression of one additional ISR-associated protein⁴ was also found to be elevated. Therefore, and as suggested by the reviewer, we now state in the discussion the possibility that the effects we observed for protein synthesis might have involved an ISR.
- Inhibition of chymotrypsin-like active sites only: Lactacystin⁵ and Epoxomicin⁶ have been reported to effectively block the chymotryptic, tryptic and peptidylglutamyl proteolytic activities of proteasomes at the concentrations used here, although the same studies also reported that inhibition of chymotryptic activities occurs at much faster rates. More recent studies, based on newer probes, confirmed that Epoxomicin (1 to 10 μ M, one hour exposures) inhibits the three β subunits (β 5, β 2, β 1) associated with these three activities quite effectively and with equal affinities⁷. Interestingly, the same study indicated that in brain tissue, β subunits associated with tryptic and chymotryptic activities are the most prevalent subunits. Finally, detailed mutation analysis (in yeast) suggests that β 5 subunits (associated with chymotryptic-like activity) strongly dominate substrate degradation *in vivo*⁸.

³ Harding et al., (2003) An integrated stress response regulates amino acid metabolism and resistance to oxidative stress. *Mol Cell* 11:619-633

⁴ Young & Wek (2016) Upstream open reading frames differentially regulate gene-specific translation in the Integrated Stress Response. *J. Biol. Chem.* pii: jbc.R116.733899.

⁵ Fenteany et al. (1995) Inhibition of proteasome activities and subunit-specific amino-terminal threonine modification by lactacystin. *Science* 268: 726–31; Craiu et al (1997) Lactacystin and clasto-lactacystin b-lactone modify multiple proteasome b-subunits and inhibit intracellular protein degradation and major histocompatibility complex class I antigen presentation. *J Biol Chem* 272:13437-13445;

⁶ Meng et al., (1999) Epoxomicin, a potent and selective proteasome inhibitor, exhibits in vivo antiinflammatory activity. *Proc Natl Acad Sci U S A.* 96:10403-10408

⁷ Berkers et al., (2007) Profiling proteasome activity in tissue with fluorescent probes. *Mol. Pharmaceutics*, 4:739–748.

⁸ Jäger et al., (1999). Proteasome beta-type subunits: unequal roles of propeptides in core particle maturation and a hierarchy of active site function. *J. Mol Biol.* 291:997-1013

Thus, to the best of our understanding, the possibility that the lack of strong effects reflects an exclusive inhibition of chymotrypsin-like activities is not very likely.

- UFD substrates – the reviewer is right. This was a mistake in our reading of the paper of Dantuma et al., (2000). Mention of UFD substrates was removed.
- Paper length - the Results and Discussion sections were shortened by about two pages, by tightening the wording throughout and moving the derivation of the mathematical framework of Fig. 8 to the Appendix.

Corresponding Author Name: Noam E Ziv

Manuscript Number: EMBOJ-2015-93594R